# Multiscale modeling of presynaptic dynamics from molecular to mesoscale

**Jonathan W. Garcia**[1,2¤]\*, **Thomas M. Bartol**[2], **Terrence J. Sejnowski**[1,2]

**1** Division of Biological Sciences, University of California San Diego, La Jolla, California, United States of America, **2** Computational Neurobiology Laboratory, Salk Institute for Biological Studies, La Jolla, California, United States of America

¤ Current address: Independent Researcher
* soiens24@gmail.com

**Data Availability Statement:** All data files and code for generating all figures in the manuscript and Supporting information have been uploaded to a publicly accessible repository on GitHub at the

## Abstract

Chemical synapses exhibit a diverse array of internal mechanisms that affect the dynamics of transmission efficacy. Many of these processes, such as release of neurotransmitter and vesicle recycling, depend strongly on activity-dependent influx and accumulation of $Ca^{2+}$. To model how each of these processes may affect the processing of information in neural circuits, and how their dysfunction may lead to disease states, requires a computationally efficient modelling framework, capable of generating accurate phenomenology without incurring a heavy computational cost per synapse. Constructing a phenomenologically realistic model requires the precise characterization of the timing and probability of neurotransmitter release. Difficulties arise in that functional forms of instantaneous release rate can be difficult to extract from noisy data without running many thousands of trials, and in biophysical synapses, facilitation of per-vesicle release probability is confounded by depletion. To overcome this, we obtained traces of free $Ca^{2+}$ concentration in response to various action potential stimulus trains from a molecular MCell model of a hippocampal Schaffer collateral axon. $Ca^{2+}$ sensors were placed at varying distance from a voltage-dependent calcium channel (VDCC) cluster, and $Ca^{2+}$ was buffered by calbindin. Then, using the calcium traces to drive deterministic state vector models of synaptotagmin 1 and 7 (Syt-1/7), which respectively mediate synchronous and asynchronous release in excitatory hippocampal synapses, we obtained high-resolution profiles of instantaneous release rate, to which we applied functional fits. Synchronous vesicle release occurred predominantly within half a micron of the source of spike-evoked $Ca^{2+}$ influx, while asynchronous release occurred more consistently at all distances. Both fast and slow mechanisms exhibited multi-exponential release rate curves, whose magnitudes decayed exponentially with distance from the $Ca^{2+}$ source. Profile parameters facilitate on different time scales according to a single, general facilitation function. These functional descriptions lay the groundwork for efficient mesoscale modelling of vesicular release dynamics.

following URL: https://github.com/soiens24/Presynaptic_Release_Rates. The MCell/CellBlender project file for the MCell model of the Schaffer collateral axon used in this paper is available at the MCell model repository at the following URL: http://mcell.cnl.salk.edu/models/presynaptic-dynamics-2022-1/.

**Funding:** This work was supported by Howard Hughes Medical Institute (https://www.hhmi.org/), by The Swartz Foundation (http://www.theswartzfoundation.org/), and by the National Science Foundation (NSF DBI-1707356 NeuroNex, https://www.nsf.org/) to TJS. MCell development is supported by the NIGMS-funded (P41GM103712) National Center for Multiscale Modeling of Biological Systems (MMBioS, https://mmbios.pitt.edu/) to TMB. The funders had no role in study design, data collection and analysis, decision to publish, or preparation of the manuscript.

**Competing interests:** The authors have declared that no competing interests exist.

## Author summary

Most information transmission between neurons in the brain occurs via release of neurotransmitter from synaptic vesicles. In response to a presynaptic spike, calcium influx at the active zone of a synapse can trigger the release of neurotransmitter with a certain probability. These stochastic release events may occur immediately after a spike or with some delay. As calcium accumulates from one spike to the next, the probability of release may increase (facilitate) for subsequent spikes. This process, known as short-term plasticity, transforms the spiking code to a release code, underlying much of the brain's information processing. In this paper, we use an accurate, detailed model of presynaptic molecular physiology to characterize these processes at high precision in response to various spike trains. We then apply model reduction to the results to obtain a phenomenological model of release timing, probability, and facilitation, which can perform as accurately as the molecular model but with far less computational cost. This mesoscale model of spike-evoked release and facilitation helps to bridge the gap between microscale molecular dynamics and macroscale information processing in neural circuits. It can thus benefit large scale modelling of neural circuits, biologically inspired machine learning models, and the design of neuromorphic chips.

## Introduction

Chemical synapses constitute the primary means of direct communication between neurons throughout the nervous system [1]. Neurotransmitters are stored in synaptic vesicles, which are docked to the plasma membrane of the axon terminal by soluble N-ethylmaieimide-sensitive factor attachment protein receptor (SNARE) complexes. Vesicle-membrane-bound synaptobrevin (v-SNARE) and the target-membrane-bound syntaxin and SNAP-25 (t-SNAREs) form energetic SNAREpin complexes where the α-helices of the v-SNAREs entwine with those of the t-SNAREs [2–4]. Synaptotagmin (Syt) proteins embedded in both membranes associate with the SNARE complex and act as $Ca^{2+}$-sensitive triggers for vesicle fusion. When the action potential of the presynaptic neuron reaches the axon terminal, it triggers a sudden influx of $Ca^{2+}$ through voltage-dependent $Ca^{2+}$ channels (VDCCs), and when a sufficient number $Ca^{2+}$ ions binds to the $C_2$ domains of synaptotagmin, it undergoes a conformational change that triggers the associated SNAREpin to zipper completely, causing the vesicle to fuse with the membrane and to release its neurotransmitter through the newly opened fusion pore [4,5] (see Fig 1).

Very often, discussion of the activity in a network tends to focus on the action potentials (spikes) and subthreshold fluctuations in membrane potential [10–12]. The utility of these measurements, however, depends on the relevance of the spike code to neural information processing. How neurons integrate their inputs and generate signals in the context of larger neural circuits largely determines the sorts of computations that the network can perform [13,14]. Biological neural networks need to represent information in a way that confers behavioral utility, but since so much of the information in the environment is irrelevant to survival, synapses may not be optimized to transmit all information faithfully, but rather selectively.

Significantly, neurons do not directly see the spiking activity of their neighbors at chemical synapses, but only detect presynaptic activation upon the release of neurotransmitter, which is a stochastic process [15]. Synapses form the basis for learning and information processing, and short-term plasticity (STP) defines a transformation from a spiking code to a neurotransmitter release code. All spiking activity is filtered through the dynamics of probabilistic synaptic

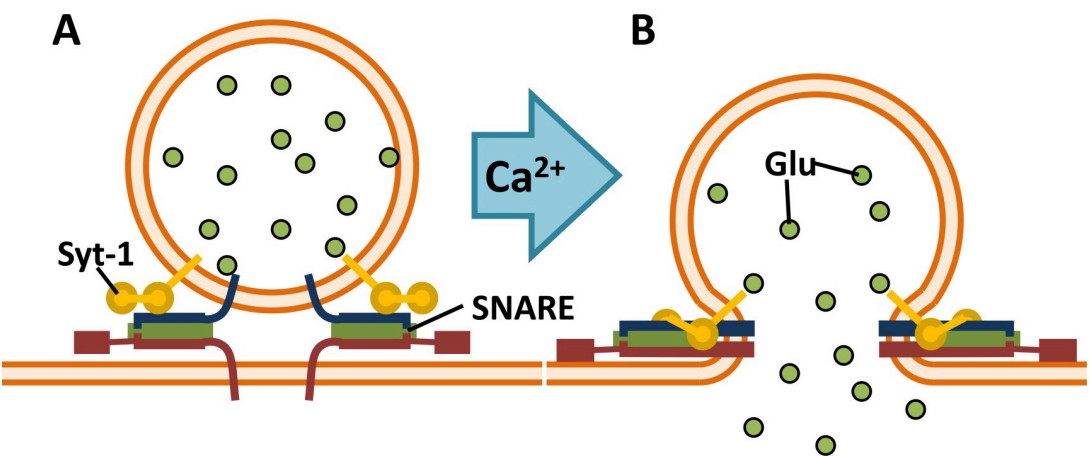

**Fig 1. SNARE Complex Structure and Dynamics.** (A) SNAREpins prior to vesicle fusion. (B) Binding of $Ca^{2+}$ to synaptotagmin (Syt-1 here, Syt-7 attaches to target membrane [6,7]) triggers full zippering of SNARE complex and, in turn, vesicle fusion [8,9].

release before the rest of the network can see it. This implies that one must first have an accurate model of release dynamics in order to understand the true nature of information processing of brain circuits. Such a model could, for instance, provide a crucial preprocessing step of motor cortex for training BCI-based prosthetics [16], or it could enable more accurate computation of the information capacity of sensory cortex by studying the "language" that neurons actually receive rather than simply the output that they generate [13,17,18].

Synaptic dysfunction has been implicated in numerous psychological disorders, including schizophrenia [19,20], bipolar disorder [19], ASD [21], and fragile X syndrome [22]. To ascertain exactly what role synapses play and what specific mechanisms might be causing or exacerbating these diseases, controlled experiments would need to be performed on the brain circuits of interest, testing which changes to synaptic function might push the network into a pathological state. Doing this in humans would pose significant problems, both technical and ethical. However, with a computational model that exhibits sufficient realism and scalability, such experiments become possible in large simulated networks, which could provide important insight into what sorts of targeted therapies to explore for treating these diseases.

The molecular simulator MCell can track the kinetics and interactions of thousands of molecules and ions in a three-dimensional model of the synapse, achieving a high degree of realism and elucidating how complex bimolecular systems may function in the absence of experimental interventions [23–27].When properly constrained by experimental data, MCell will not only automatically reproduce observed features such as asynchronous vesicle release, facilitation, and depression in the probability of release [25], but it can also make surprising predictions that are later confirmed through experiment [28,29]. However, it quickly becomes too computationally expensive to scale up to the many synapses that exist even in relatively simple neural circuits.

The ultimate goal, therefore, is to develop a presynaptic model that captures realistic phenomenology while maintaining computational scalability. To that end, in this paper we develop a mathematical model that describes the phenomenology of presynaptic dynamics, using the MCell model of [25] as a reasonable approximation to ground truth. Our simplified model takes an arbitrary spike train and uses it to approximate the single-vesicle release rate histograms that would emerge from running an infinite number of MCell simulations, taking into account the facilitation in release rate of both synchronous and asynchronous release

dynamics. The results are easily amenable to event-driven models, producing the generating distributions for temporally asynchronous vesicle release times with respect to arbitrary sequences of action potentials. Thus, it provides an avenue for large-scale simulations of spiking neural networks with many realistically performing synapses without incurring high computational costs, enabling investigations into how various presynaptic mechanisms can affect computation at the circuit level.

## Results

### $Ca^{2+}$-evoked dynamics of vesicle release

Diffusion plays a key role in presynaptic processes. Simplified models of $Ca^{2+}$-dependent presynaptic dynamics may assume that the axon terminal is locally well mixed, equivalent to saying that diffusion happens infinitely fast, at least relative to the spatial and temporal scales being studied. However, MCell allows one to add a spatial component to molecular simulations, which can account for certain phenomena that well mixed molecular kinetics models cannot capture [23,24,26]. To characterize the process of $Ca^{2+}$-dependent neurotransmitter release, we based two models on the presynaptic model of [25]: a spatially explicit model implemented in MCell, and an equivalent "well-mixed" model. Both models contained voltage-dependent $Ca^{2+}$ channels (VDCCs) that let $Ca^{2+}$ ions into the presynaptic volume in response to an action potential stimulus, a calbindin (CB) buffer that moderated diffusion of the ions via its binding kinetics, and plasma membrane $Ca^{2+}$-ATPase (PMCA) pumps that helped intracellular $Ca^{2+}$ concentration return to equilibrium over time (see Fig A in S1 Text for state transition diagrams). However, the two models differed in that the MCell model relied heavily on diffusion of molecular species through space and discrete state transition events occurring over time, while the well-mixed model treated all molecular interactions as occurring within the same point in space and tracked continuous state probabilities over time (see Methods).

Comparing these models, we found that the diffusion of $Ca^{2+}$ and calbindin through the axonal volume affects both the timing and the probability of spike-evoked vesicle release, depending on the distance from the $Ca^{2+}$ source. Fig 2 compares the well-mixed simulation without diffusion to the equivalent MCell simulations performed at multiple distances from the VDCC $Ca^{2+}$ source. The shape of the $Ca^{2+}$ transient measured in MCell displays marked qualitative differences from that obtained without diffusion: $Ca^{2+}$ sensors near the VDCC source see a much higher peak concentration with an extra component of decay immediately following the peak; those farther away progressively lose the fast peak until nothing is left but an extremely small distance-independent component. The extra component of the proximal $Ca^{2+}$ curve, which does not appear in the well-mixed model, likely arises from local saturation in the nanodomains near the VDCC cluster, where the very high free $[Ca^{2+}]_i$ temporarily saturates both the calbindin buffer and the PMCA pumps [30]. Farther out, the MCell model qualitatively matches the well-mixed model more closely, until at very large distances, the fast components almost completely disappear. The distance-independent components represent a sustained global elevation in $[Ca^{2+}]_i$ that persists due to the excess $Ca^{2+}$ that has yet to unbind from the calbindin buffer. The slowest component has a magnitude comparable to resting $[Ca^{2+}]_i$ and a time constant of around 1 second.

One would expect the strength of spike-evoked neurotransmitter release to diminish with increasing distance from the $Ca^{2+}$ source, where $Ca^{2+}$ has more time to diffuse and bind to buffer molecules before reaching the sensor. In fact, numerous studies have found that vesicles of the readily releasable pool (RRP) fall into one of two subpopulations, depending on their physical location of vesicles within the synapse: vesicles located very near $Ca^{2+}$ channels release

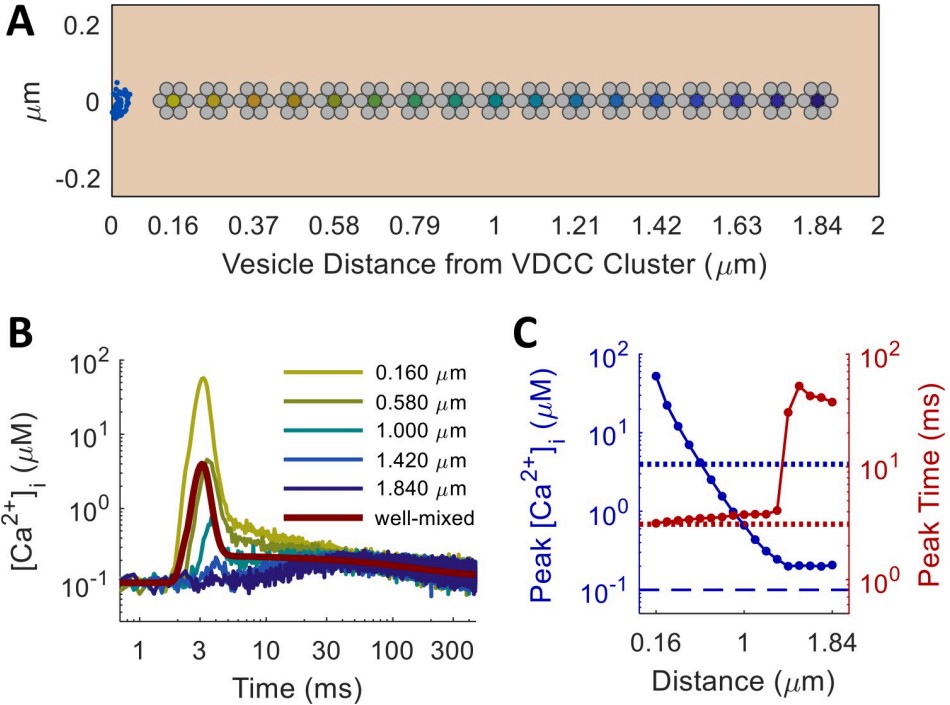

**Fig 2. Spatial Modeling of Spike-Evoked Ca²⁺ Transients.** (A) Ca²⁺ sensors (dark yellow through dark blue filled circles) at vesicle cluster centers, displaced linearly from cluster of Ca²⁺ channels (blue half-disk on the left); distance in µm, $d_n = 0.160 + 0.105n$ for $n \in \{0, ..., 16\}$. (B) $[Ca^{2+}]_i$ measured over time in MCell (dark yellow through dark blue) and in the deterministic well-mixed model (maroon). MCell traces averaged from 2000 trials of MCell simulations with $\Delta t = 0.1$ ms. Color transitions from yellow for vesicles proximal to the VDCC Ca²⁺ source to blue for vesicles far away, as in A. Proximally (distally) measured $[Ca^{2+}]_i$ displays more (fewer) components of decay than are evident in the deterministic model. (C) Logarithmic plots of peak $[Ca^{2+}]_i$ (blue) and peak time (red) as a function of distance from Ca²⁺ source; peak $[Ca^{2+}]_i$ drops off exponentially with distance from VDCC cluster; amplitude of latent Ca²⁺ dominates over the initial action-potential-evoked influx after 1.4 µm.

quickly in response to spikes, while those farther away are more reluctant [31,32]. To explore how the release rate profiles vary with distance, we established a linear array of Ca²⁺ sensors along the length of the model axon, with a cluster of 50 VDCCs arranged in a half-disk at one end (Fig 2A). Reflective boundaries on the ends of a 2-µm tube effectively simulated the effects of having one cluster of 100 VDCCs every 4 µm, consistent with previous models of the Schaffer collateral axon [25,33]. Running the model again for 2000 trials, with a single action potential stimulus applied at the beginning, we obtained Ca²⁺ traces measured at each point along the axon. For the first 1.4 µm, free Ca²⁺ from the initial influx dominated, and the peak concentration declined exponentially with distance (length constant 0.204 µm; Fig 2C). Farther out, global accumulation and depletion of Ca²⁺ dominates, which, although spike-evoked, does not vary in magnitude with distance and acts over a much longer time scale and at a much lower level than most of the spike-triggered Ca²⁺.

Running these simulations in MCell, rather than as a much simpler well-mixed model, was essential for capturing both distance-dependent effects and temporal features of the Ca²⁺ waveform. The well-mixed assumption, which ignores diffusion and treated all chemical processes as occurring at the same point in space, does not hold at the spatial and temporal scales of interest in the synapse [34,35]. As seen in Fig 2C, peak Ca²⁺ dropped precipitously even over fractions of a micron away from the VDCC cluster, and the shape of the response changed dramatically over this same scale, transitioning from a predominantly synchronous to a

predominantly asynchronous profile, even before the $Ca^{2+}$ sensors started responding. These trends, elucidated by the spatial MCell model, were completely absent in the space-less well-mixed simulation (Fig 2B, maroon line), even when all other aspects of the model remained the same, such as the number of VDCCs, calbindin buffer molecules, and PMCA pumps and the set of all state transitions for each molecular species (see Methods for details). Note also from Fig 2B that the transition in time from the fast synchronous component to the extended asynchronous component was much sharper in the case without space. The extra $Ca^{2+}$ decay component arose from local saturation effects. After the initial rapid influx, the calbindin buffer immediately around the VDCC cluster became saturated, causing the high free $Ca^{2+}$ that remains to overwhelm the PMCA pumps' ability to evacuate it from the area. The pumps removed it at a constant maximum rate, leading to a short linear decay only evident very near the VDCCs (yellow traces, Fig D panel A in S1 Text) or when all calbindin is removed from the simulation (Fig E in S1 Text). Such effects did not appear in the well-mixed case because all buffer molecules and pumps were simultaneously available to all the free $Ca^{2+}$, preventing any local saturation from occurring. Thus, in light of all these effects, the spatial MCell model was crucial for the task of properly characterizing the $Ca^{2+}$ transient in the synapse.

Furthermore, diffusion of $Ca^{2+}$ through the cytoplasm depends in part on the presence of cytoskeleton components and the vesicles themselves [36–39], which can obstruct diffusion. For the purposes of this investigation, we included just the vesicles surrounding each release site, arranged in hexagonal clusters of seven and centered over the locations described in Fig 2A, each vesicle measuring 35 nm in diameter [25], to act as obstacles. Cytoskeletal components, such as the actin filaments and microtubules abundant in the axon [40], can have a substantial slowing effect on the diffusion rate of cytoplasmic proteins [37] such as calbindin (CB). To account for the higher viscosity of cytoplasm, the diffusion constant for calcium used in this paper ($D_{Ca}$ = 220 $\mu m^2$/s) is several times slower than the free diffusion of $Ca^{2+}$ in water. Our simulations explicitly account for the interactions of diffusing calcium with diffusing CB and stationary calcium binding proteins, resulting in an effective diffusion constant of around 50 $\mu m^2$/s, consistent with experimental measurements of the apparent diffusion constant of $Ca^{2+}$ in cytoplasm [25,41]. Furthermore, the diffusion constant of CB in our MCell simulations ($D_{CB}$ = 28 $\mu m^2$/s, [25,42]) also takes into account the viscosity of cytoplasm, falling between the infinitely fast diffusion of well-mixed models and the completely immobile conditions in other investigations of synaptic calcium transients [38,39]. We found that CB does not contribute significantly toward transporting $Ca^{2+}$ ions to the active zone, acting instead primarily to slow and suppress free calcium influx, shortening the initial transient and extending the time window in which $Ca^{2+}$ is available for binding to the SNARE complex (see Fig E in S1 Text).

Although many more proteins are involved in coordinating release kinetics at active zones [4,8,9,32], for validation purposes we restrict the scope of this paper to the function of synaptotagmins. The model of synaptotagmin-mediated release used in our simulations followed the dual $Ca^{2+}$-sensor model of Sun et al. [5], which includes mechanisms for both fast/synchronous and slow/asynchronous release. In excitatory hippocampal synapses, these synchronous and asynchronous modes of release may correspond to the roles of synaptotagmin-1 (Syt-1) and synaptotagmin-7 (Syt-7), respectively [43]. The model incorporates cooperative binding of $Ca^{2+}$ to multiple sites on the sensor, requiring five $Ca^{2+}$ ions before triggering synchronous release and two $Ca^{2+}$ ions for asynchronous release (see Fig 3). Because both binding and unbinding rates for the synchronous mechanism are substantially higher than those for the asynchronous mechanism, Syt-1 produces rapid release over a very narrow window relative to spike arrival time, while Syt-7 produces slow release over a much more extended window. Table 1 contains the values used in this model for $Ca^{2+}$-binding and unbinding rates with each release mechanism, along with the rates of vesicle fusion from the fully bound states ($\gamma_S$ and

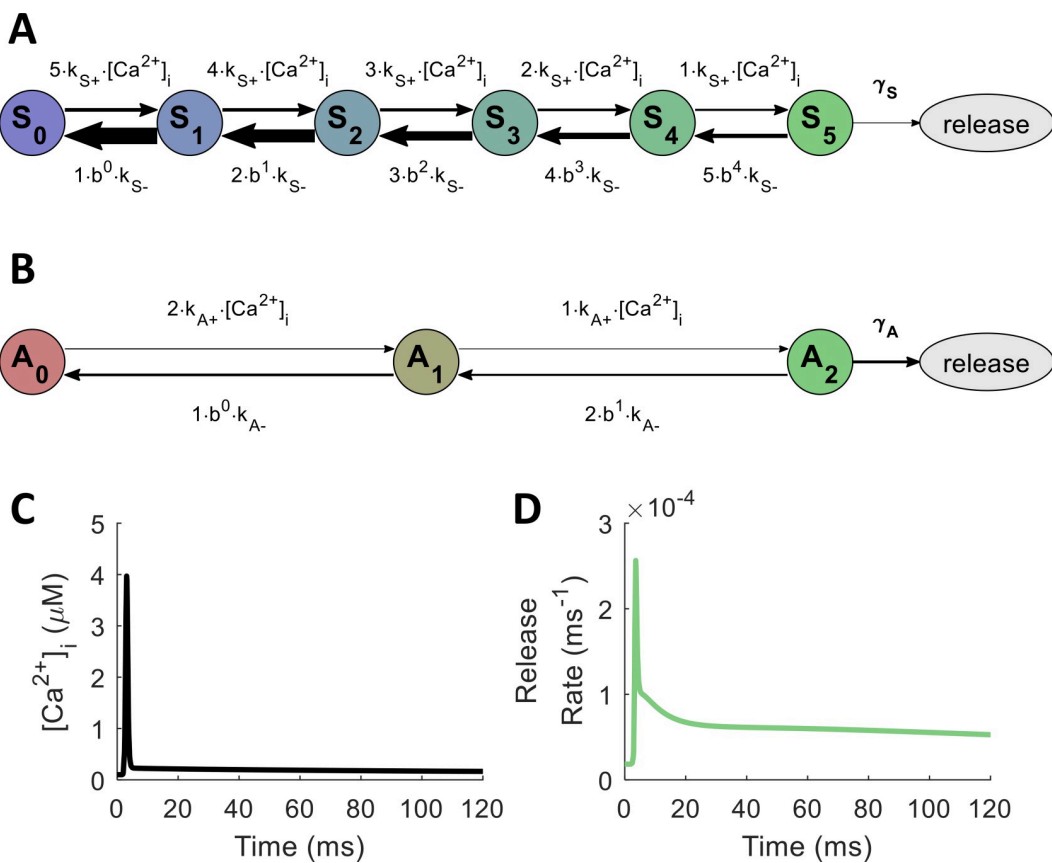

**Fig 3. Model of Ca²⁺-Evoked, Synaptotagmin-Mediated Neurotransmitter Release.** (A,B) Model adapted from Sun et al. [5]. $\gamma_S$ and $\gamma_A$ represent rates of vesicle fusion from the releasable states of the synchronous and asynchronous mechanisms, respectively. (A) Ca²⁺-bound states for Syt-1 (synchronous release); $S_n$ indicates n Ca²⁺ ions bound to the synchronous release mechanism. (B) Ca²⁺-bound states for Syt-7 (asynchronous release); $A_n$ indicates n Ca²⁺ ions bound to the asynchronous release mechanism. (C,D) Action-potential-like stimulus delivered to model axon starting at 0 ms. Diffusion is assumed to be instantaneous, and molecular state probabilities are tracked deterministically over time. (C) Free [Ca²⁺]ᵢ in response to single action potential. (D) Instantaneous vesicle release rate in response to buffered Ca²⁺ from both synaptotagmin-mediated release mechanisms.

$\gamma_A$) and the time constant for the post-release refractory period ($\varepsilon$) [44,45], which features in the Nadkarni et al. [25] MCell model.

The MCell model, because it tracks thousands of individual particles through Markov chain Monte Carlo simulations [23,24,26], can both capture very realistic synaptic dynamics and uncover their underlying molecular causes, which would be difficult to obtain through other methods. Unfortunately, this realism can also obscure the patterns necessary for building simplified models. First, many processes, such as asynchronous or "mini" release events [46,47],

**Table 1. SNARE Release State Transition Parameters.**

|  | synchronous |  | asynchronous |  | other parameters |
|---|---|---|---|---|---|
| $k_{S+}$ | $6.12 \times 10^7$ M⁻¹s⁻¹ | $k_{A+}$ | $3.82 \times 10^6$ M⁻¹s⁻¹ | $b$ | 0.25 |
| $k_{S-}$ | $2.32 \times 10^3$ s⁻¹ | $k_{A-}$ | 13 s⁻¹ | $\varepsilon$ | 6.34 ms |
| $\gamma_S$ | $6.0 \times 10^3$ s⁻¹ | $\gamma_A$ | 50 s⁻¹ |  |  |

Values taken from Nadkarni et al. [25], adapted from Sun et al. [5].

occur slowly enough that many thousands or millions of simulated trials would be required to uncover precise functional descriptions, which could become computationally prohibitive. For instance, the histograms of synchronous release obtained from 2000 trials of MCell in Fig 4 offer little information on spontaneous release from the Syt-1 mechanism between action potentials, and synchronous release far from the VDCC cluster (blue) hardly occurs at all. Second, the fact that vesicles deplete upon release hides how the instantaneous single-vesicle release rate actually changes with time. The tails of the release distributions fall off too quickly as vesicles are removed from the simulation over time, and any paired-pulse facilitation (PPF) in single-vesicle release probability is countered by the release-dependent depletion in the model (Fig 4D).

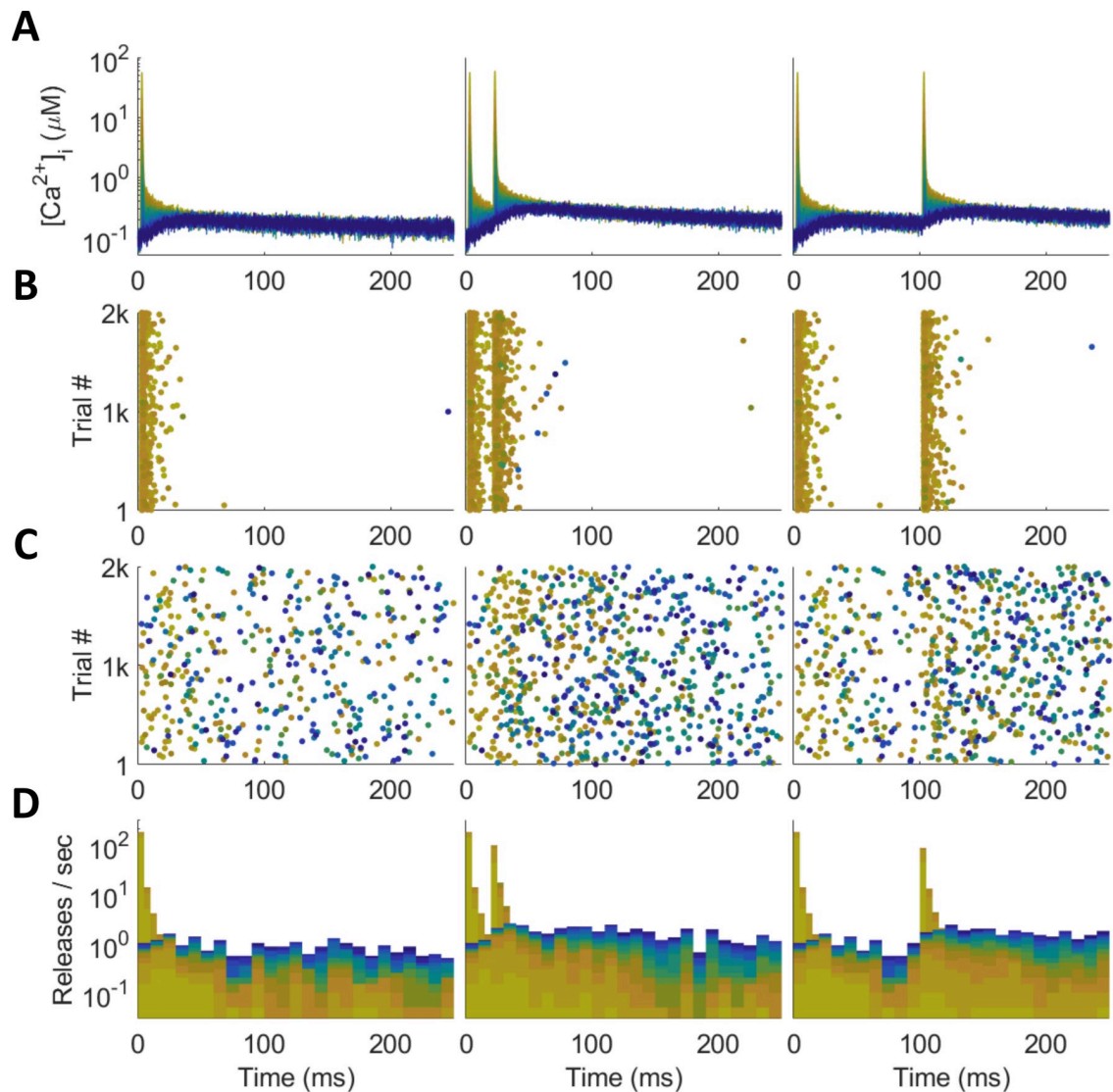

**Fig 4. Synchronous and Asynchronous Release in MCell.** Color indicates distance from VDCC source, with yellow representing a nearby Ca²⁺ sensor and dark blue a distant one (as in Fig 2A and 2B). Action-potential-like stimulus delivered at 0 ms (left), followed by another at 20 ms (center) and 100 ms (right). (A) Spike-evoked Ca²⁺ traces that drive release. (B) Synchronous release raster. (C) Asynchronous release raster. (D) Synchronous (tall, thin bars) and asynchronous (short, wide bars) release stacked histogram. Most synchronous releases happen close to the Ca²⁺ source; asynchronous releases distributed across all distances.

The only way to avoid these depletion effects in MCell would be to run many millions of trials with a single vesicle to track how the vesicle's alacrity for release fluctuates with the $Ca^{2+}$ history detected at its position. For these reasons, we decided not to depend on the release histograms generated by many trials of MCell for building a phenomenological model. Instead, we used the $Ca^{2+}$ traces generated by MCell, which do not suffer from the aforementioned problems, to drive deterministic simulations of the SNARE state probability dynamics (as described in Methods), effectively producing what an infinite number of trials would produce in MCell with the same $Ca^{2+}$ data. Thus, using the deterministic release rates driven by the stochastic MCell $Ca^{2+}$ data balances the necessary realism of MCell with the smoothness and insights required for designing a versatile phenomenological model.

## Reducing from molecular simulations to phenomenological model

By driving a deterministic simulation of SNARE dynamics with the $[Ca^{2+}]_i$ waveform obtained from MCell, one can see that each release mechanism induces vesicle fusion with a histogram that essentially follows a multi-exponential form (Fig 5). The release rate profiles ($r(t)$, where $r \in \{S, A\}$ may refer to synchronous or asynchronous release rate) rise quickly from baseline after the spike and decay with several exponential components, approximated as

$$r(t) = r_0 + \sum_{c=1}^{N} \frac{P_c}{\tau_c} e^{-t/\tau_c} u(t), \qquad (1)$$

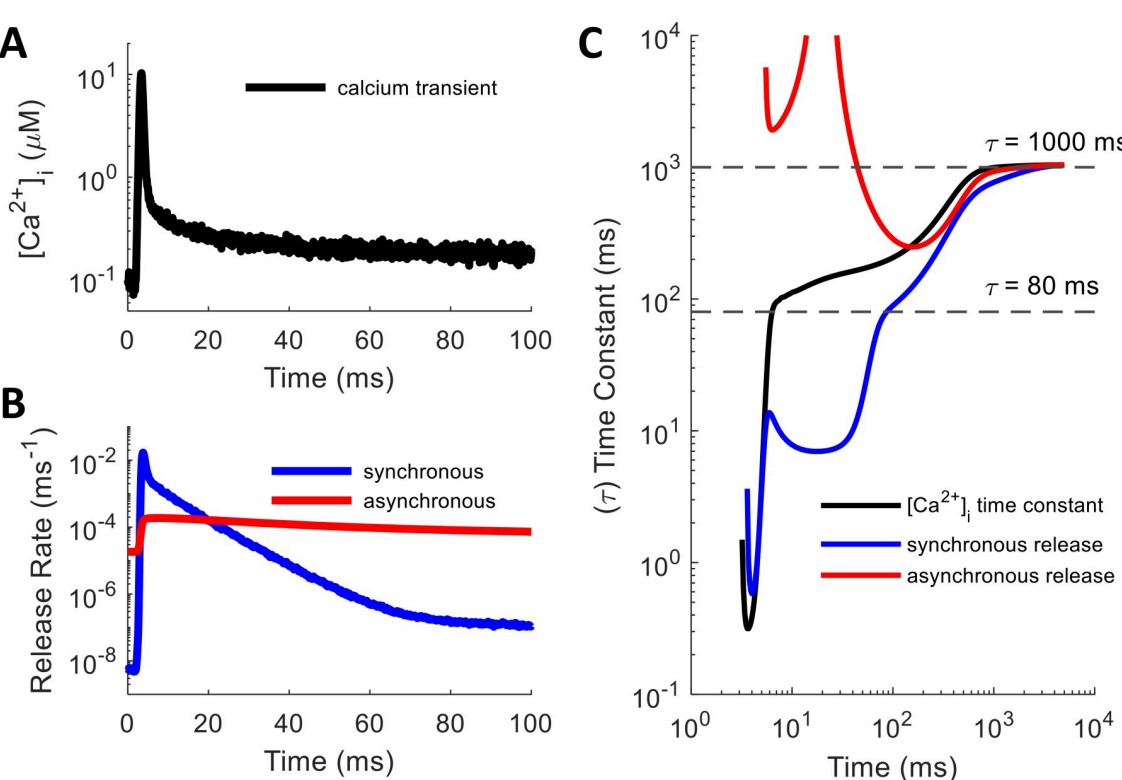

**Fig 5. Multi-Exponential Shape of $Ca^{2+}$-Driven Vesicle Release Rate.** (A,B) Plots given as semi-log to highlight exponential decay components (straight line segments of profiles). (A) A single, spike-evoked $[Ca^{2+}]_i$ transient, which drives (B) the synchronous and asynchronous release rates. (C) Instantaneous time constants for $Ca^{2+}$, synchronous, and asynchronous curves, calculated from the well-mixed model (see Eq (2)). Long release rate time constants (around 80 ms and 1000 ms; dashed lines) follow $Ca^{2+}$ curve due to slow un-buffering of latent $Ca^{2+}$. Asynchronous starts high because fast and slow components have comparable magnitude and become conflated; it goes up to infinity where additive effects cause the curve to flatten.

where $r_0$ is the spontaneous release rate (related to "mini"-EPSCs [47,48]; $S_0 = 5.70\times10^{-9}$ms$^{-1}$; $A_0 = 1.84\times10^{-5}$ms$^{-1}$), $t$ is the time since the last spike, u($t$) is the Heaviside step function (so that release occurs only for $t\geq0$), $N$ is the number of exponential decay components, $\tau_c$ are the time constants of exponential decay, and $P_c$ are the expected number of releases from each component for a single vesicle. Note that because the release rate profile is not a probability distribution, but rather it represents the instantaneous rate of release conditioned on having not released yet (see Methods), its integral $P_c$ can potentially exceed one (just as the integral over the spontaneous component $r_0$ for $t = 0...\infty$ is infinite). The probability that the exponential component causes release at any point in time, ignoring the other components, is $p_{rc} = 1-\exp(-P_c)$ (and the integrated release probability for the spontaneous component is $p_{r0} = 1 -\exp(-\infty) = 1$). The existence of multiple exponential components is apparent from the linear segments visible in log-linear space for synchronous and asynchronous release rates in Fig 5B. To calculate the time constants of exponential decay (Fig 5C), we used the slope of the logarithm of the release rate curve according to

$$\tau(t) = -\left(\frac{\mathrm{d}}{\mathrm{d}t}[\ln(r^*(t) - r^*(0))]\right)^{-1}, \tag{2}$$

where $\tau(t)$ is the instantaneous time constant and $r^*(t)$ is the observed instantaneous release rate. We used the well-mixed model for the derivative calculation because it had no noise in the release rate profiles.

Most of the Ca$^{2+}$ that enters the axon following an action potential quickly binds with the calbindin buffer before diffusing to the SNARE complex, causing a narrow spike in the free [Ca$^{2+}$]$_i$ available to the release mechanism. Therefore, most of the spike-evoked release occurs in response to this narrow window of influx. To test how each release mechanism responds to transient Ca$^{2+}$ spikes, we supplied an instantaneous burst of Ca$^{2+}$ to a single time step of the deterministic model, allowing us to measure the impulse-response function. These simulations were repeated for various resting Ca$^{2+}$ levels ([Ca$^{2+}$]$_{i0}$), ranging from 0 to 10 μM, to see how the presence of Ca$^{2+}$ at equilibrium affects the response to spike-evoked transients. As Fig 6 shows, when there is no resting [Ca$^{2+}$]$_i$, the rate of release for both synchronous and asynchronous mechanisms rises quickly in response to a sudden influx before dropping exponentially with a single exponential component (black). However, when [Ca$^{2+}$]$_{i0}$ settles at some level greater than zero, an extra exponential component emerges for both mechanisms (blue and red lines). The exponential decay time constants seem to be mostly independent of resting [Ca$^{2+}$]$_{i0}$ at low levels, but they drop off more quickly as spontaneous release rates begin to overtake the spike-evoked rates at high concentrations. The extra component emerges as a result of the back-and-forth Ca$^{2+}$-binding and unbinding processes, where finite baseline [Ca$^{2+}$]$_{i0}$ likely provides a "floor" to "bounce off of" in terms of the number of Ca$^{2+}$ ions bound to the release mechanism. Note, however, that even though it depends on equilibrium [Ca$^{2+}$]$_{i0}$, this secondary release component is still purely spike-evoked and arises due to the nonlinearity of the system. The baseline rate $r^*(0)$ was subtracted off to ensure that the function approached zero prior to taking the logarithm (dotted lines). Time constants in Fig 6C were calculated using Eq (2).

From the above, it would seem that each mechanism should have three components to its release histogram: a constant spontaneous rate that increases with [Ca$^{2+}$]$_{i0}$, a fast exponential component that acts in response to an impulse of spike-evoked Ca$^{2+}$, and a slower spike-evoked component that results from a "rebound" interaction with the Ca$^{2+}$ floor. However, the profiles of the release rate histograms display more complexity than this, which will be discussed in more detail below. Significantly, [Ca$^{2+}$]$_i$ does not drop instantly to baseline after the

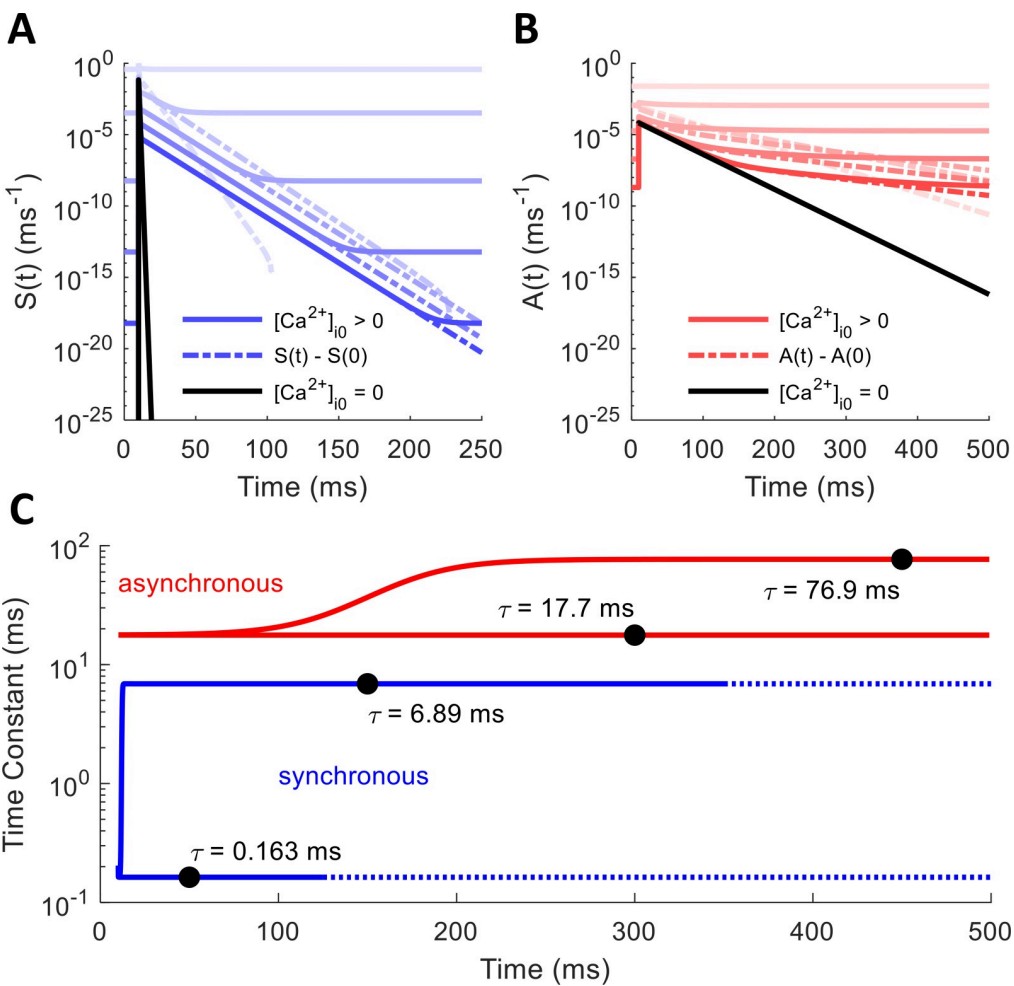

**Fig 6. Synchronous and Asynchronous Release Rates in Response to Ca$^{2+}$ Impulse at Different Resting Concentrations.** Instantaneous impulse of Ca$^{2+}$ delivered at 10 ms. Solid lines represent true release rate; dotted lines have spontaneous rates subtracted off to show secondary exponential components. Black lines show release rate decaying with a single exponential component with no baseline [Ca$^{2+}$]$_i$. For other curves, [Ca$^{2+}$]$_{i0}$ ranges from 0.001 μM to 10 μM. (A) Synchronous release rate over time: S(t). (B) Asynchronous release rate over time: A(t). (C) Instantaneous release rate decay time constants for synchronous and asynchronous mechanisms. Fast components (lower blue and red lines) determined from profiles with [Ca$^{2+}$]$_{i0}$ = 0 (black lines in A and B). Slower components (upper blue and red curves) determined from cases with small [Ca$^{2+}$]$_{i0}$.

initial influx, but some leftover Ca$^{2+}$ continues to have a small effect over a long time window as it slowly unbinds from the calbindin buffer (see Fig E in S1 Text). This allows a small but noticeably enhanced rate of release efficacy to continue out to hundreds or thousands of milliseconds before returning fully to baseline (within noise). Fig 5C shows the effect that this latent Ca$^{2+}$ has on producing longer time constants in the decay of the release rate profiles, using the smooth curves obtained from the well-mixed model.

Of course, neurotransmitter release cannot begin at exactly the moment of the spike, both because the action potential itself is not an instantaneous process and because it takes finite time for Ca$^{2+}$ to diffuse from the VDCC source, through the buffer, to the Ca$^{2+}$ sensor in the SNARE complex. MCell represents this complex process with a Markov chain Monte Carlo method (MCMC). Because of this, the release process cannot begin until the spike-evoked Ca$^{2+}$ arrives, which time may vary randomly relative to the timing of the spike. Thus, the process of buffered

diffusion acts as a temporal filter on the release dynamics, transforming the equation of release to

$$r(t) = r_0 + \sum_{c=1}^{N} \frac{P_c}{\tau_c} \left( e^{-t/\tau_c} u(t) \right) * a(t; k_c, \mu_c, \sigma_c),$$

(3)

where $a(\cdot)$ is the temporal filter and $k_c$, $\mu_c$, and $\sigma_c$ are parameters to be discussed below. The convolution operation effectively smears the start time of the average release profile in a way that accounts for random temporal jitter across trials.

Importantly, the release-start-time filter $a(\cdot)$ must integrate to one over all real numbers. That way, it does not affect the probability of release, only its timing. The temporal filter chosen is an ex-Gaussian distribution, resulting from the convolution of an exponential distribution of rate $k$ with a normal distribution of mean $\mu$ and standard deviation $\sigma$:

$$a(t; k, \mu, \sigma) = (ke^{-kt}u(t)) * \left( \frac{1}{\sigma\sqrt{2\pi}} e^{-\frac{(t-\mu)^2}{2\sigma^2}} \right)$$

$$= \int_{-\infty}^{t} ke^{-k(t-t')} \frac{1}{\sigma\sqrt{2\pi}} e^{-\frac{(t'-\mu)^2}{2\sigma^2}} dt'$$

$$= ke^{-k\left(t - \left(\mu + \frac{\sigma^2}{2}k\right)\right)} \Phi\left( \frac{t - (\mu + \sigma^2 k)}{\sigma} \right),$$

(4)

where $\Phi(\cdot)$ represents the CDF of the zero-mean, unit-variance normal distribution. In the limit where $\sigma \to 0$, this CDF simply becomes the shifted Heaviside step function $u(t-\mu)$, and $a(t; k, \mu, \sigma) \to ke^{-k(t-\mu)}u(t - \mu)$, which is just a rightward shift in time of the exponential distribution by $\mu$. The values of $\mu$ and $\sigma$ result from the sum of the delays caused by numerous random processes, including the timing of $Ca^{2+}$ entry relative to the spike, the accumulation of collision events during Brownian motion, and the binding/unbinding events with the calbindin buffer and SNARE complex. Assuming that the individual events of the buffered diffusion process are numerous and similar enough for a given spike, the central limit theorem states that the sum of their delays should approximate a normal distribution [49]. The value of $k$ represents the rate of some limiting step in the process of buffered diffusion, and it slows with increasing distance between the VDCC source and the $Ca^{2+}$ sensor in the SNARE complex. Keep in mind that these parameters constitute only a phenomenological approximation to the exact filter, but they work well enough for the purposes of this paper.

For an event-driven model, which this paper is working towards, the convolutional operation can be implemented by sampling a normally distributed random number (with mean and standard deviation $\mu_c$ and $\sigma_c$) and an exponentially distributed random number (with rate $k_c$) and adding them to the spike time to determine the start time for the release response. In other words, combine the spike time with an ex-Gaussian random delay to determine when the release component begins to respond to the spike, following Eq (1). In aggregate, across many trials with the same spike time, the release histogram will approach Eq (3).

With the mathematical description of the release rate profiles in mind, we ran a fitting algorithm (see Methods) to determine the values of the parameters for each profile. Initially, we used release profiles driven by $Ca^{2+}$ measured at 400 nm from the cluster of 100 VDCCs, which provides a physiologically realistic probability of release for a single vesicle (around 0.04) [50]. The synchronous release mechanism exhibits more exponential decay components in its release rate histogram than does the asynchronous mechanism (4 versus 3), likely because it has more $Ca^{2+}$ binding sites (5 versus 2) and because it operates on a faster time scale. Fig 7 shows how the fitted parametric release profiles match the simulated release

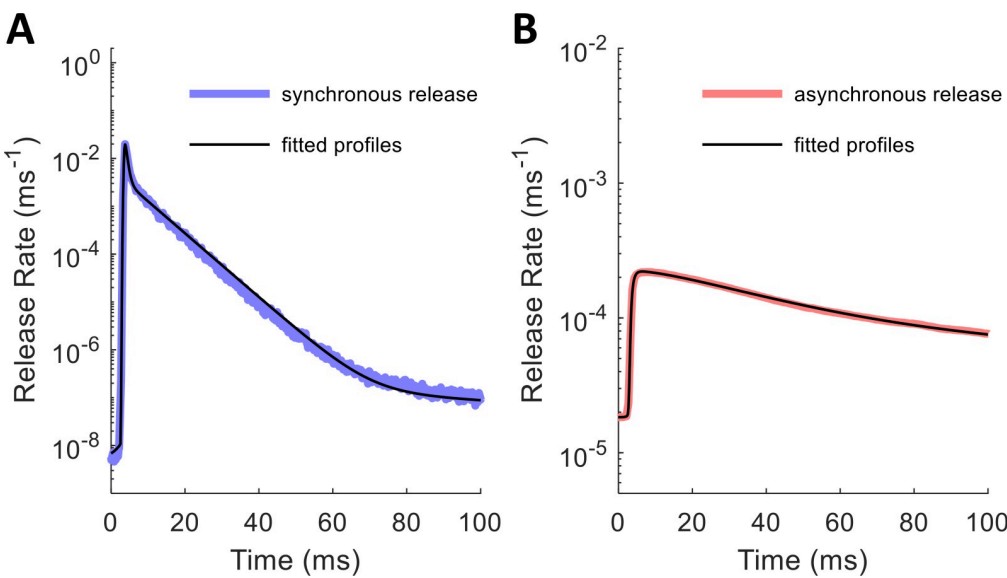

**Fig 7. Fitted Release Rate Histogram Profiles for a Single Spike.** Parameter values given in Table 2. (A) Synchronous release rate: true histogram (blue) with estimated histogram (black). (B) Asynchronous release rate: true histogram (red) with estimated histogram (black).

profiles to within noise across multiple orders of magnitude; the noise in the simulated profiles was due to fluctuations in $[Ca^{2+}]$ at the $Ca^{2+}$ sensor. Table 2 lists the values of the best-fit profile parameters.

Keep in mind that the $\mu$ values are somewhat arbitrary in that they depend on exactly when during the action potential that the spike time is taken to occur. Action potential waveforms last a couple of milliseconds (see Fig B in S1 Text) [51]; the values for $\mu$ above used a point on the action potential waveform immediately prior to the rising phase as the spike time. Using the peak of the action potential would take away about 2 ms from all values of $\mu$. Again, the time point along the action potential where the spike is counted is arbitrary, but it must be consistent across all components.

## Combining release profiles for multiple spikes

Having established the shape of the release profile for a single spike, we considered how the release profiles of multiple spikes in a train would combine. Consider first the case of two

**Table 2. Spike-Evoked Release Rate Parameters.**

| component | P | τ | k | μ | σ |
|---|---|---|---|---|---|
| $S_1$ | 0.0175 | 0.163 ms | 1.79 ms$^{-1}$ | 3.41 ms | 0.168 ms |
| $S_2$ | 0.0220 | 6.50 ms | 18.0 ms$^{-1}$ | 3.56 ms | 0.0977 ms |
| $S_3$ | $1.70\times10^{-5}$ | 80.0 ms | 0.526 ms$^{-1}$ | 10.0 ms | 4.44 ms |
| $S_4$ | $1.10\times10^{-5}$ | 1000 ms | 0.142 ms$^{-1}$ | 50.0 ms | 11.5 ms |
| $A_1$ | $3.72\times10^{-3}$ | 17.7 ms | 1.60 ms$^{-1}$ | 3.05 ms | 0.243 ms |
| $A_2$ | 0.0111 | 76.9 ms | 0.0759 ms$^{-1}$ | 4.00 ms | 1.14 ms |
| $A_3$ | 0.0136 | 1000 ms | 0.0337 ms$^{-1}$ | 76.5 ms | 21.9 ms |

Parameter values calculated for a single spike following a period of low activity. Valid for $Ca^{2+}$-sensitive synchronous and asynchronous release mechanisms located 400 nm from a cluster of 100 VDCCs.

spike times, $t_{s1}$ and $t_{s2}$ (where $t_{s2} > t_{s1}$). After the first spike, the release mechanism will respond to a $Ca^{2+}$ influx after some delay with the release profile of Eq (1) shifted in time by $t_{Ca1}$, where the delay between the spike time and the arrival of the $Ca^{2+}$ influx is distributed according to the ex-Gaussian temporal delay filter:

$$t_{Ca1} - t_{s1} \sim a(t - t_{s1}; k, \mu, \sigma). \tag{5}$$

Fig 8A–8C shows visually how this temporal delay filter affects a given release profile component.

When the second spike arrives at the release site, the VDCCs produce another influx of $Ca^{2+}$ that can again propagate to the SNARE complex, building on the $Ca^{2+}$ from the first spike. The buffered diffusion again involves an ex-Gaussian-distributed delay, after which the

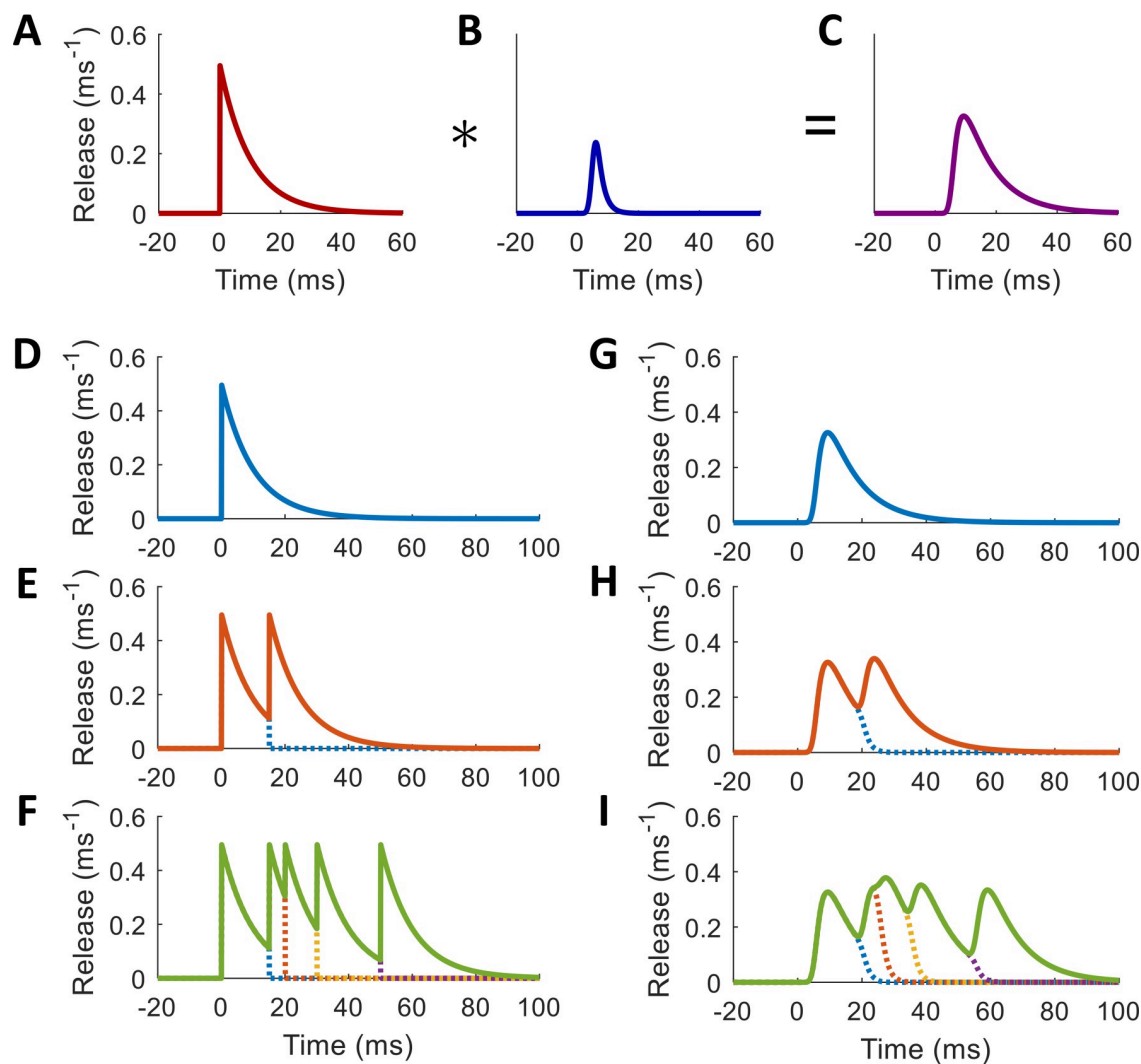

**Fig 8. Convolutional Filter Applied to a Component of a Release Rate Function.** Toy model with P = 5, τ = 10 ms, k = 0.5 ms⁻¹, μ = 5 ms, and σ = 1 ms. (A) Unfiltered release rate component. (B) MCMC ex-Gaussian filter shape. (C) Filtered release profile produced by convolving the release rate profile with the temporal delay filter. (D-F) Release rates in response to spike trains without applying delay filter. (G-I) Release rates in response to spike trains with delay filter applied. (D,G) Response to one spike. (E,H) Response to two spikes. (F,I) Response to multiple spikes. Dotted lines show how the histogram of response to one spike falls off with interference from the response to the following spike. Spike times at 0, 15, 20, 30, and 50 ms.

release mechanism starts responding to the second spike at time $t_{Ca2}$. With a future event-driven simulator in mind, we treated the arrival of $Ca^{2+}$ from the second spike as a transition point between release-time generating functions. That is, the synapse stops generating release times in response to the first spike (whose release profile was shifted by $t_{Ca1}$) and starts generating release times in response to the second spike (with a release profile shifted by $t_{Ca2}$), according to

$$r(t; \{t_{Ca1}, t_{Ca2}\}) = r_0 + \sum_{c=1}^{N} \frac{P_c}{\tau_c} e^{-(t-t_{Ca1})/\tau_c} \, \mathrm{u}(t - t_{Ca1})(1 - \mathrm{u}(t - t_{Ca2}))$$

$$+ \sum_{c=1}^{N} \frac{P_c}{\tau_c} e^{-(t-t_{Ca2})/\tau_c} \, \mathrm{u}(t - t_{Ca2})$$

$$= r(t - t_{Ca1})(1 - \mathrm{u}(t - t_{Ca2})) + r(t - t_{Ca2}), \tag{6}$$

where r(t) is the unfiltered release profile from Eq (1). Notice that Eq (6) does not account for facilitation yet. In this section, we focus on the interaction of individual spikes' release profiles, reserving the discussion of facilitation for the next section. For a spike train with an arbitrary number of spikes, this becomes

$$r(t; T_{Ca}) = \sum_{t_{Cai} \in T_{Ca}} \left( r(t - t_{Cai}) \prod_{\substack{t_{Caj} \in T_{Ca} \\ t_{Caj} > t_{Cai}}} (1 - \mathrm{u}(t - t_{Caj})) \right), \tag{7}$$

where $T_{Ca} = \{t_{Ca1}, t_{Ca2}, \ldots\}$ is the set of all $Ca^{2+}$ arrival times, each resulting from the combination of a spike time with an ex-Gaussian-distributed delay. Fig 8D–8F shows what these profiles should look like for a certain set of parameters in response to various spike trains.

While the above formulation relies on fixed delay times, an event-driven simulator will need to sample new delay times on every trial for a given spike train, as well as different delay times for each release component. Therefore, we calculated the probability $D_c(t)$ that the $Ca^{2+}$ sensor has begun responding to the $Ca^{2+}$ from the latest spike for release component c, allowing for a gradual switch from one spike-evoked release profile to the next, taking into account the variation of delay among all possible trials. For two spike times, $t_{s1} < t_{s2}$, the net release profile for each component of Eq (6) becomes

$$r_c(t; \{t_{s1}, t_{s2}\}) = r_c(t - t_{s1})(1 - D_c(t - t_{s2})) + r_c(t - t_{s2}). \tag{8}$$

In other words, the response to the first spike is cut short after the second spike by $D_c(\cdot)$ to give way to the new release response. And every time another spike arrives, it decreases the probability of release relative to the first spike multiplicatively, such that each component of Eq (7) becomes

$$r_c(t; T_S) = \sum_{t_{si} \in T_S} \left( r_c(t - t_{si}) \prod_{\substack{t_{sj} \in T_S \\ t_{sj} > t_{si}}} (1 - D_c(t - t_{sj})) \right), \tag{9}$$

where $T_S = \{t_{s1}, t_{s2}, \ldots\}$ is now the set of all spike times.

Because this formulation now uses spike times rather than delayed $Ca^{2+}$ arrival times, the step functions $\mathrm{u}(t-t_{Caj})$ of Eq (6) and (7) have been smeared out in time by the temporal delay filter of Eq (4) to become $D_c(t-t_{sj})$. Assuming the latest spike arrives at t = 0, $D_c(t)$ is simply the

cumulative distribution of the temporal delay filter:

$$D_c(t) = \int_{-\infty}^{t} a(t; k_c, \mu_c, \sigma_c) \mathrm{d}t$$

$$= \Phi\left(\frac{t - \mu_c}{\sigma_c}\right) - e^{-k_c\left(t - \left(\mu_c + \frac{\sigma_c^2}{2}k_c\right)\right)} \Phi\left(\frac{t - (\mu_c + \sigma_c^2 k_c)}{\sigma_c}\right). \tag{10}$$

More intuitively, by letting $\sigma_c \rightarrow 0$, the Gaussian component becomes a delta function, and the first-release distribution function above becomes much more simply

$$D_c(t) = (1 - e^{-k_c(t - \mu_c)})\mathrm{u}(t - \mu_c). \tag{11}$$

Thus, after the second spike, the histogram of releases from the first spike drops off exponentially, while those due to the second spike rise and fall as for the first spike. When $\sigma_c$ is small relative to the median interspike interval, third spikes have an almost imperceptible effect at cutting the first profile short relative to the second spike's effect. Fig 8G–8I shows how this transition works in response to the same spike trains as in Fig 8D–8F.

## Characterizing facilitation in vesicle release rates

The discussion above has focused on the release response of a single vesicle with a constant probability of release across spikes. However, many synapses display a facilitation in release probability from one spike to another [14,44,52–54]. This results both from an accumulation of $Ca^{2+}$ in the presynaptic space [52] and from a stochastic accumulation of $Ca^{2+}$ on the sensor (Syt) of the SNARE complex. Simulations with the MCell model demonstrate how nonlinear binding cooperativity in the $Ca^{2+}$ sensors induces facilitation in excess of what would be expected from cytoplasmic $Ca^{2+}$ buildup alone (Fig I in S1 Text). This happens because on some trials, $Ca^{2+}$ accumulates on the sensor, not enough to trigger vesicle fusion on the first spike, but enough to increase the probability of reaching the releasable state after subsequent spikes. As can be seen in Fig H in S1 Text, $Ca^{2+}$ entry from one spike can predispose the distribution of bound states of the sensor to trigger release with greater alacrity on subsequent spikes.

Furthermore, the level of facilitation depends to some extent on the full history of spiking activity in the synapse. In the simplest case, the probability of release on one spike should depend only on the probability for the previous spike and on the time since the previous spike. However, the level of facilitation is not a simple function of the most recent activity but depends on the rate of stimulation prior to the last spike. To explore the space of facilitation dynamics more fully, we applied spike trains with spike ramps of different rates and durations, to see how quickly facilitation builds up, followed by single probe spikes at increasing interspike intervals (ISI), to see how quickly it decays back to baseline (see Methods). Fig 9 shows examples of how these different spike trains affect the rates of synchronous and asynchronous release.

Facilitation does not affect all components of release equally. Therefore, we derived a general facilitation function $F_c(\cdot)$ that affects each release component $c$ independently. The area under the curve of each component of the release rate profile (see Eq (1) and (3)) depends on the facilitation factor according to

$$P_c(n) = P_{c0} \cdot F_c(n), \tag{12}$$

where $P_{c0}$ is the baseline value and $n$ is the index of the current spike. To ensure that the function works for arbitrary spike trains, the factor $F_c(n)$ needs both to grow somehow from spike

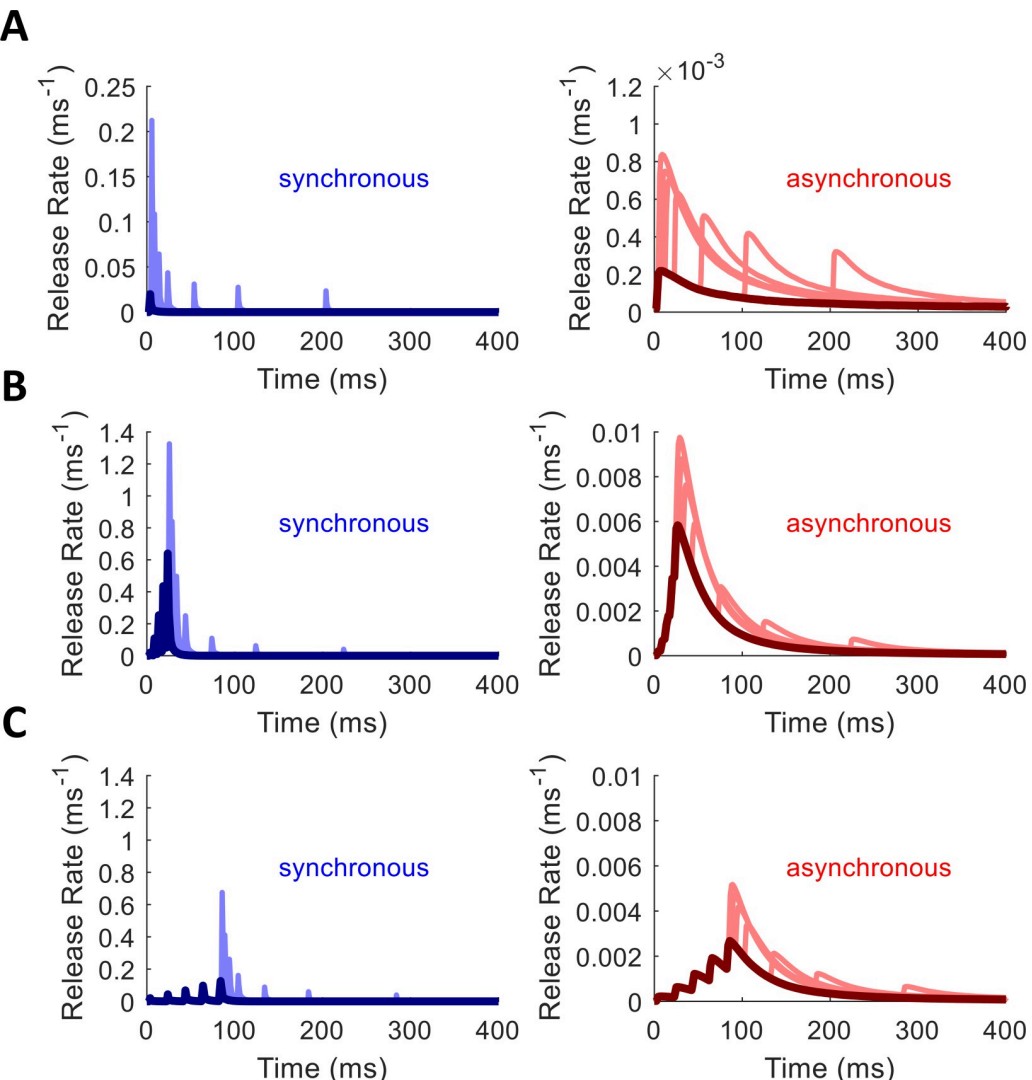

**Fig 9. Empirical Facilitation in Synchronous and Asynchronous Release Rates.** Release rate profiles facilitate in response to single spikes (A) or to spike ramps (B,C). Probe spikes of increasing ISI reveal how facilitation then decays back toward baseline after a delay. Each rise in release rate is triggered by a spike event. Synchronous release rate profiles shown in blue. Asynchronous profiles shown in red. Profiles from multiple runs are overlaid in each panel. Dark colors represent response to initial spike or spike ramp (common to all traces on a plot). Light colors represent profiles from different runs in response to single probe spikes at different ISIs following the initial spike or ramp. (A) After single spike, paired-pulse facilitation decays with increasing ISI (probe ISIs of 2, 5, 10, 20, 50, 100, 200 ms). (B) 5-spike ramp with a 5-ms ISI shows strong facilitation in release rate (dark colors) followed by rapid decay seen at the probe spikes (light colors). (C) 5-spike ramp with a 20-ms ISI shows weaker facilitation in the ramp phase but a similar rate of decay at the probe spikes. Note the orders of magnitude difference in scale between synchronous and asynchronous release rates, as well as the change in scale from (A) to (B,C).

to spike and to decay back toward one for large ISIs. This growth can happen in a highly non-linear fashion, so to account for this, we take $F_c(n)$ to be a nonlinear combination of linear facilitation factors $f_{ci}(n)$ such that

$$F_c(n) = \prod_{i=1}^{M_c} f_{ci}(n)^{\xi_{ci}}, \qquad (13)$$

where $M_c$ represents the number of facilitation components (either one or two for all functions

explored below), and $\xi_{ci}$ represents the nonlinearity applied to facilitation component $i$ of release component $c$. Each $f_{ci}(n)$ accounts for some aspect of the internal state of the SNARE complex, in terms of how the expected number of $Ca^{2+}$ ions bound changes with time, that helps determine the probability of release on subsequent spikes.

In the simplest case, each $f_{ci}(n)$ would decay exponentially from its previous value $f_{ci}(n-1)$ before incrementing by one:

$$f_{ci}(n) = f_{ci}(n-1)e^{-\Delta t/\tau_{ci}} + 1, \tag{14}$$

where $\Delta t$ is the delay from the previous spike to the current one. The increment of one is meant to account for the influx of about the same amount of $Ca^{2+}$ during each action potential. This formulation ensures that even after infinitely long intervals, the facilitation factor will equal a value no less than $F_c(0) = 1$, allowing the release components to return to their baseline values of $P_c = P_{c0}$ during long periods of inactivity, as expected.

However, this formula implies that for an infinitely fast rate of stimulation, $f_{ci}(n)$ could grow toward infinity, producing an infinitely fast rate of release, all of which are impossible. More realistically, there should exist some finite saturation level, $L_{ci}$, such that the facilitation function could never theoretically exceed

$$F_c(\infty) = \prod_{i=1}^{M_c} L_{ci}. \tag{15}$$

The value of this upper limit is constrained by the rates of vesicle fusion from the fully bound states of the SNARE complex ($\gamma_S$ and $\gamma_A$ in Table 1) and by the maximum level of $Ca^{2+}$ buildup in the presynaptic space. When facilitation is still well below this level, it should continue to increment by approximately one on every spike, but this increment should fall to zero quickly enough that facilitation never exceeds saturation. Setting a maximum number of equal-sized steps to saturation for each component, $N_{ci} = L_{ci}^{1/\xi_{ci}}$, the value of $f_{ci}(n)$ becomes

$$f_{ci}(n) = f_{ci}(n-1)e^{-\Delta t/\tau_{ci}} + 1 - \left(\frac{f_{ci}(n-1)e^{-\Delta t/\tau_{ci}}}{N_{ci}}\right)^{N_{ci}}. \tag{16}$$

The new term subtracted off at the end ensures that $f_{ci}(n)$ never exceeds $N_{ci}$, just as $Ca^{2+}$ cannot accumulate to infinite concentrations but is limited by the electrochemical gradient across the cell membrane [55]. An alternative would be simply to set $f_{ci}(n) = N_{ci}$ whenever a step size of one would cause it to exceed this limit, but the formula in Eq (16) allows for a smoother approach.

With the model for facilitation established, we sought to fit it to the empirical changes observed in release rate for complex spike trains. For simplicity, we took facilitation to apply only to the $P_c$ parameters, which control for the magnitude of each release component, although in principle the parameters of the temporal filter ($k_c$, $\mu_c$, and $\sigma_c$) might also increase ($\xi_{ci}>0$) or decrease ($\xi_{ci}<0$) with spike history. As discussed in Methods, we explored 136 unique spike trains for how both spike rate (along the spike ramp) and ISI (of the probe spike) affect the release rate in response to the last spike.

For the fitting algorithm, we used a simplex method for gradient descent, since the derivatives of the error function are difficult to compute (see Methods). The values of the $P_c$ parameters were allowed to vary within bounds, while the profile time constants and the temporal filter parameters were held constant. The best-fit set of values for $P_c$ were found for the spike-response profile at the end of each spike train, after which the meta-parameters of the facilitation functions could be fitted to the patterns in $P_c$. Fig 10A shows some examples of fitted release rate functions fit to baseline (blue) and facilitated (yellow) profiles. Fig 10B depicts how

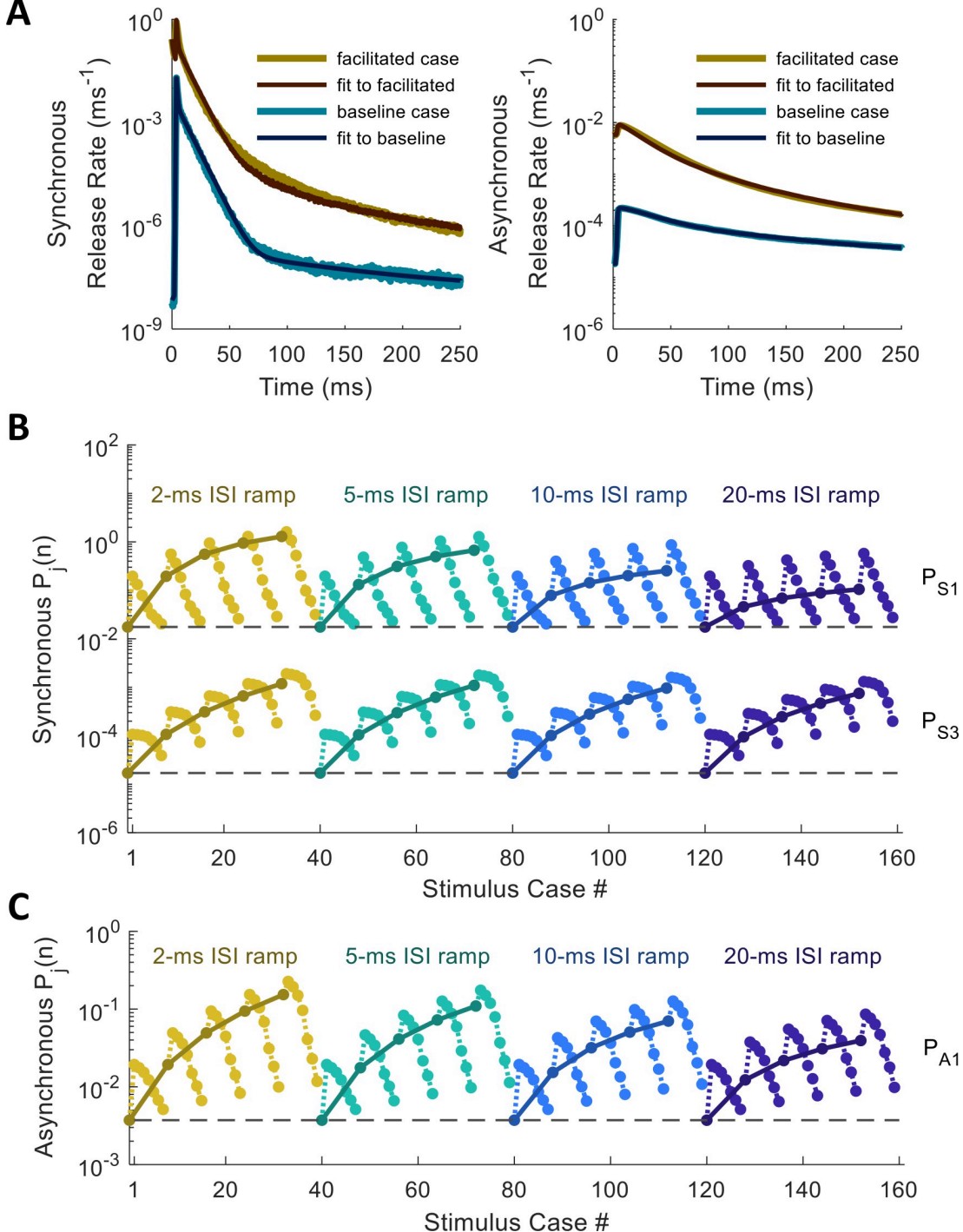

**Fig 10. Release Rate Parameters and Facilitation Metaparameters Fitted to Empirical Histogram Profiles.** (A) Synchronous and asynchronous profiles fitted for baseline (un-facilitated) case, and for highly facilitated case (probe spike 5 ms after 5-spike ramp of 5-ms ISIs). (B,C) Release fidelity values fitted case-by-case (dark colors) overlaid with predictions from best-fit facilitation functions (light colors) for synchronous (B) and asynchronous (C) components.

**Table 3. Metaparameters for Facilitation of Release Fidelity.**

| component | $P_0$ | $\tau$ | $N$ | $\xi$ | $L = N^{\xi}$ |
|---|---|---|---|---|---|
| $S_{11}$ | 0.0175 | 95.9 ms | 7.00 | 1.27 | 11.8 |
| $S_{12}$ | | 7.66 ms | 2.32 | 2.93 | 11.8 |
| $S_{21}$ | 0.0220 | 13.1 ms | 10.0 | 1.23 | 17.0 |
| $S_{22}$ | | 114 ms | 17.6 | 1.68 | 125 |
| $S_3$ | $1.70 \times 10^{-5}$ | 199 ms | 12.5 | 2.67 | 846 |
| $S_4$ | $1.10 \times 10^{-5}$ | – | 1 | 0 | 1 |
| $A_{11}$ | $3.72 \times 10^{-3}$ | 141 ms | 12.2 | 1.48 | 40.0 |
| $A_{12}$ | | 17.2 ms | 12.5 | 0.996 | 12.4 |
| $A_2$ | 0.0111 | 126 ms | 12.1 | 1.67 | 64.4 |
| $A_3$ | 0.0136 | – | 1 | 0 | 1 |

First column shows baseline magnitudes of integrated release rate, duplicated from Table 2. First and second components of synchronous release and first component of asynchronous release facilitate with two time constants each. Smallest component of both release mechanisms does not facilitate. $P_0$ is integrated release rate for the un-facilitated case (baseline), τ is the time constant of decay for each facilitation component, N is the number of linear facilitation steps to saturation, ξ is the nonlinearity parameter, and L is the maximum facilitation factor contributed by each component. Valid for $Ca^{2+}$-sensitive synchronous and asynchronous release mechanisms located 400 nm from a cluster of 100 VDCCs.

release magnitudes varied for the release magnitude components (first ($P_{S1}$) and third ($P_{S3}$) components of synchronous release and first component ($P_{A1}$) of asynchronous release) across all 136 facilitation spike trains (see Fig J in S1 Text for a depiction of how stimulus cases are ordered).

Through trial-and-error, we found that the two components of synchronous release with the fastest time constants, along with the fastest component of asynchronous release, each required two facilitation components to explain their patterns of change from case to case. The synchronous and asynchronous release components with "medium" time constants (each close to 80 ms) could each be fitted with a single facilitation component. The slowest release components, with time constants of 1000 ms due to latent $[Ca^{2+}]_i$ released from the buffer (see Fig 5C), were constrained not to facilitate, since changes in these components seemed to have a negligible effect on fitting error. Table 3 records the facilitation meta-parameters obtained from the fits, along with the baseline values for release fidelity for each component ($P_0$). During each step of the fitting algorithm, these metaparameters were used to generate predictions for the state of facilitation across all 136 spike-train cases, and error was calculated as the fraction of the variance of the "true" release fidelity values unexplained by the predicted pattern (Fig L in S1 Text; see Methods).

## Discussion

### Advantages and limitations of treating MCell as ground truth

Basing the new model on MCell has distinct advantages over biological experiments in terms of both flexibility and precision when it comes to model validation. As an example, estimates of $[Ca^{2+}]_i$ in neurons obtained from fluorescent reporters in physiological experiments may provide accurate estimates of slow (tens of milliseconds) $Ca^{2+}$ transients [56,57], but the buffering kinetics of the calcium reporters can act as a low-pass filter, obscuring the faster (0–5 milliseconds) components of $Ca^{2+}$ dynamics [27]. Molecular simulations like MCell, on the other hand, can capture these fast transients, since they track every particle, which may be crucial for correctly modeling fast, $Ca^{2+}$-dependent synaptic processes like synchronous vesicular release of neurotransmitter [48].

Furthermore, neurotransmitter release may occur asynchronously with respect to the arrival time of action potentials, following some time-dependent distribution [48]. Experimental methods for determining release rate would offer far less control of presynaptic conditions over the number of trials that would be required to tease out the same resolution of detail as is possible with controlled simulations. Therefore, we chose to constrain ourselves to validating the model developed in this paper against an MCell model, which has itself been validated already against hippocampal Schaffer collateral synapses [25].

However, this approach is limited on several levels. First, it assumes that MCell can correctly replicate the dynamics of diffusion and molecular interactions of biophysical systems through its Markov chain Monte Carlo framework [23,24,26]. Second, it assumes that the molecular kinetics of the included species match their true kinetics. Finally, it assumes that the molecular species and biological systems modelled are the only ones present in the presynaptic compartment, or at least that any other systems would produce only negligible changes to the phenomenology of the synapse. MCell has been sufficiently well validated to satisfy the first assumption [26], at least for the temporal and spatial scales of interest here (hundreds of microseconds and several microns).

The second assumption is valid insofar as the molecular models used by MCell correctly represent reality, both in terms of the molecular state diagrams and in terms of the binding and interaction kinetics reported by other groups: Sun et al. [5] for the descriptions of $Ca^{2+}$-driven SNARE kinetics for vesicular release; Bischofberger et al. [51] for the VDCC dynamics for spike-evoked $Ca^{2+}$ influx; Nägerl et al. [58] for the high- and medium-affinity sites of the calbindin $Ca^{2+}$ buffer; and Sneyd et al. [59] for the kinetics of the PMCA pumps. The state diagrams and kinetic parameters for these species are summarized in Fig A and Table A in S1 Text. Simplifying assumptions inevitably go into models such as these, which limit the accuracy of any model based on them. However, for the purposes of this paper, we assume that these models reproduce experimental results sufficiently well to use them.

The greatest limitations to model accuracy come from the third assumption in that the true variety of systems and molecular species in biological synapses far exceeds what MCell represents [60]. For instance, the Nadkarni et al. [25] model did not include any endoplasmic reticulum (ER), which stores intracellular $Ca^{2+}$ and has a significant effect on neuronal signaling, nor did it include the ryanodine receptors (RyR) and inositol 1,4,5-trisphosphate receptors (IP3R) that unleash these $Ca^{2+}$ stores [61]. Inclusion of such an ER would likely alter the shape of the $Ca^{2+}$ transient and increase the probability of neurotransmitter release, possibly over longer time scales, as in synaptic augmentation [50,62]. Another system that would significantly affect $Ca^{2+}$ dynamics over multiple spikes is the facilitation and inactivation of $Ca^{2+}$ channels mediated by $Ca^{2+}$-calmodulin (CaM) and $Ca^{2+}$ binding proteins (CaBP) [46,63]. These interactions might help control release-independent depression (RID) and the frequency-dependent recovery (FDR) from depression [53,64] by restricting $Ca^{2+}$ influx over extended spike trains. Furthermore, the presynaptic $Ca^{2+}$ buffer includes more than just calbindin [65,66] and diffusion is limited by the plethora of intracellular microstructures [36–39], beyond just the synaptic vesicles included in this study. Because location and movement through space plays a crucial role in phenomenology (see Fig 2 and Figs D, F, and G in S1 Text), investigating how these features affect release dynamics would require rerunning the MCell model with them included.

The advantages of using MCell as ground truth, we believe, outweigh the limitations enumerated above. A simulated synaptic model allows for much finer experimental control and consistency from trial to trial, while yielding far more precise results than physiological experiments. Measuring release rate, probability, facilitation, and depression at biological synapses is difficult and requires a number of problematic assumptions [67]. Using MCell allows for

precise measurements of unmodified $Ca^{2+}$ traces [27] and of single-vesicle release rates, even controlling for such confounding processes as the post-release refractory period [25]. Any new features, including arbitrary numbers of vesicle pools and synaptic processes, can easily be included in future work.

## Comparison to other models of short-term synaptic dynamics

In response to a given spike train, this model produces a release rate profile that represents the rate of a time-varying Poisson process, the average release activity of an infinite number of synapses (or trials) responding to the same spike train. The multi-exponential form of the release rate profile and the facilitation factors that modify it evolve deterministically for a given sequence of spike times. Incorporating this as a deterministic synaptic model into a spiking neural network simulation would allow for comparison with other deterministic models of short-term plasticity, such as that introduced by Tsodyks and Markram [68]. Their model uses a relatively simple representation of the utilization of synaptic resources, tracking the fraction of resources in recovered, active, and inactivated states, to flexibly model both short-term depression and facilitation [14], and has proven useful in simulated neural network contexts for producing complex behavior [69]. On the other hand, the predictive power of the Tsodyks-Markram model is limited by how it abstracts away all the internal processes of the synapse, conflating presynaptic (vesicle availability, release probability) with postsynaptic (neurotransmitter receptor saturation and desensitization) resources. In contrast, our model more closely tracks the vesicle release phenomenology that results directly from $Ca^{2+}$-evoked molecular dynamics. This allows us to explore the contribution of more fine-grained synaptic features to whole-network computations.

Another limitation of the Tsodyks-Markram model is its deterministic formulation, representing average or aggregate synaptic behavior rather than single-trial or single-synapse behavior. However, the presence of trial-to-trial stochasticity in synaptic transmission may have important implications for the learning and information processing performed in neural circuits [18,70,71]. Our model can capture such trial-to-trial and synapse-to-synapse variability if we sample release events from the time-varying Poisson process defined by the release rate profile.

An example of a presynaptic model that captures probabilistic release is that by Maass and Zador [72]. The Maass-Zador model tracks short-term facilitation in response to spike history and short-term depression in response to release history, calculating the probability of release at each spike time and generating Boolean release events according to this probability. However, these release events only occur exactly at spike times, in contrast to the spontaneous and asynchronous release that occurs in real synapses [73,74]. Furthermore, the phenomenology of the Maass-Zador model, while elegant, arises from mathematical abstractions rather than from physiologically grounded mechanisms, making it susceptible to producing unrealistic behavior and limiting its utility as a testable model.

Kandaswamy et al. [50] present another presynaptic model that aims for more physiologically grounded realism, employing multiple mechanisms of vesicle recycling, facilitation, augmentation, and release-dependent depression, each with its own set of parameters. Some parameters were constrained to values derived from earlier studies, while other parameters were fit to experimental measurements of changes in synaptic strength in response to various constant-frequency spike train stimuli. Although the resulting model does qualitatively well at matching the experimental data, it lacks generalizability. In particular, the free parameters of their model, which they adjusted to fit the experimental data, depend empirically on the frequency of stimulation but without any discernible pattern that would provide insight into

their physiologic origin (see Table 2 in [50]). Furthermore, the model for facilitation treats the first spike as a special case relative to all subsequent spikes, excluding the first spike from facilitation to fit the data. A realistic short-term facilitation model should scale the probability of release on the first spike as though it occurred in the middle of a train after an infinite ISI, since the synapse should return to its baseline state after a sufficiently long interval of no spiking activity, which our model accomplishes.

Future work with our model will involve implementing it in an event-driven framework, where release events are sampled from the time-varying Poisson process defined by the release rate profile. This approach will provide a powerful way to include a highly generalizable facilitation function and sub-millisecond vesicular release phenomenology, grounded in molecular kinetics, into a highly scalable and computationally efficient presynaptic model. In contrast to the models reviewed above, this model can achieve both the stochasticity and the asynchronicity that are characteristic of real synapses, while maintaining a clear mapping between its parameters and the underlying molecular mechanisms. This approach will be crucial for exploring the impact of different presynaptic mechanisms on the computational performance of large neural circuits.

The main advantage of our model is its balance of computational efficiency with physiologically grounded realism. It accounts for both asynchronous and spontaneous release events, which the Maass-Zador model lacks. For neural network simulations where the presence of stochasticity in synaptic transmission is more important than reproducing true dynamics, the Maass-Zador model may suffice. However, for investigations into how presynaptic mechanisms of vesicle release affect information transmission and network behavior, our model provides and indispensable layer of flexibility.

Furthermore, our model provides a highly flexible and explanatory framework. Each of the parameters for describing the profile of the release histogram ($P$, $\tau$, $k$, $\mu$, $\sigma$; see Eq (1), (3), and (4)) has an almost direct link to the underlying physiology, whether to the $Ca^{2+}$-binding and vesicle fusion kinetics of the SNARE complex ($P$ and $\tau$) or to the stochastic delay in response to the spike caused by buffered diffusion of $Ca^{2+}$ ($k$, $\mu$, and $\sigma$). Furthermore, the facilitation function has sufficient complexity to account for the changes seen in neurotransmitter release fidelity of a wide variety of spike train patterns. Importantly, all spikes are treated equally: Eq (13) and (16) apply as consistently to the first spike as to the $n$-th. While it falls short in terms of computational efficiency relative to the Kandaswamy et al. model, it makes up for it in terms of biophysical plausibility and its utility for testing hypotheses regarding synaptic function.

## Importance of facilitation function parameterization

Different synapse types in different regions of the brain employ different short-term plasticity functions, including various forms of facilitation and depression, transforming spike timing codes into neurotransmitter release timing codes [68]. We do not know what the precise computational roles of facilitation and depression play in neural circuits, but it seems likely to involve more than just high-pass or low-pass filtering of spike trains [13,17,18,75–77]. The model of facilitation presented in this paper enables the exploration of this question both because of its grounding in explainable molecular physiology via MCell and because of its amenability to efficient simulation of complex neural circuits.

One possible shortcoming of the facilitation function presented in this paper is that we did not explore the steady-state behavior in our simulations. Although the steady-state facilitation factors can be calculated from the fitted parameters, we will need to run MCell simulations with longer spike ramps to confirm their values for long spike trains of different frequencies or

to make other improvements to the facilitation model. For random spike trains of similar length to those we investigated, we expect facilitation in the detailed MCell model to follow the predictions of our phenomenological model. This is because we designed the ramp-probe spike protocol (see Methods) to extract as much information about facilitation as possible from short spike trains, such that short random spike trains should fall well within the space spanned by the protocol. However, behavior may start to diverge for longer trains, which fall outside of the space we tested. Therefore, running longer MCell simulations both to characterize steady-state behavior of facilitation and to test on long, random spike trains is a priority in the next stage of this project.

Furthermore, since the release rates explored in this paper deal with single-vesicle release probabilities, release-dependent depression (i.e. depletion-dependent depression) should occur naturally when applying the model to simulations of synapses with finite readily releasable vesicle pools. However, other forms of release-independent depression may be more relevant to natural spike frequencies [64,78] and could be incorporated into an extra facilitation factor within the current model by using a negative $\xi$ as the exponent in Eq (13). Thus, the facilitation function can be naturally extended to include release-independent depression mechanisms, although further research is necessary to confirm the kinetics of the other active zone molecules of non-hippocampal synapses that underlie such mechanisms.

## Future refinements

Because so much about release probability and timing depends on the precise magnitude and time course of the $Ca^{2+}$ transient, accurate modelling of $Ca^{2+}$ dynamics using MCell is essential to future refinements of our model. For instance, intracellular calcium stores, including ER and its associated receptors and channels [61,62,79] and mitochondria and its associated calcium uniporter [80], can impact the $Ca^{2+}$ signaling over the long term. The diffusion of $Ca^{2+}$ through the presynaptic space can be affected in ways not captured by our current model by buffering molecules other than calbindin and signaling molecules such as CaM and CaBP1 [46,63,65,66] and by the geometry of cytoskeletal microstructures [36,37]. We may also explore the effect of varying the binding kinetics and expression levels of the various calcium buffers, which can vary by both neuronal type and developmental stage [81–84], although based on our experiments with removing CB, we can predict that this would only affect the number, magnitudes, and time constants of the exponential components in the release rate profiles (see Figs D and E in S1 Text), which could be easily determined, but would not alter the qualitative shape of the profiles. Additional mechanisms that can influence the $Ca^{2+}$ transient include the shape of the presynaptic action potential, which can vary in cases such as Fragile X syndrome [85], spatially localized $Ca^{2+}$ spikes mediated by presynaptic NMDA receptors [86], and retrograde signaling via endocannabinoid receptors [87].

Other considerations of synaptic physiology, those which do not affect the shape of the spike-evoked $Ca^{2+}$ transient, can be characterized without running full MCell simulations. These include the structure and dynamics of the SNARE complex. In our model, we assumed that each vesicle employs two $Ca^{2+}$ sensors for triggering vesicle fusion, Syt-1/2 for synchronous release and Syt-7 for asynchronous release, as characterized by the Sun et al. [5] model. Each mechanism acted independently, and together they served as the sole mechanism of release. However, many more molecules comprise the SNARE complex, each affecting release fidelity in complex ways [4,8,9,32]. In fact, the inclusion of extra molecules in the SNARE assembly may play a crucial "superpriming" step in enhancing the release alacrity of already-primed vesicles [32,52]. Furthermore, although a single SNARE complex is sufficient to induce spike-evoked release [88], each vesicle may have multiple SNARE complexes associated with

it, which is necessary for fast vesicle fusion [89]. We predict that multiplying SNARE complexes would simply multiply release rate in proportion. It is less clear what effects that other molecules, such as complexins and Muncs, would have on release kinetics. A quantitative understanding of their molecular interaction kinetics is required before they can be applied to this model. Once obtained, however, we can apply deterministic simulations of state probabilities (see Methods) for this more complex SNARE model, similar to what we did in this paper. If these simulations are driven by the improved $Ca^{2+}$ transients obtained from the more mechanistically exhaustive MCell simulations described above, we can derive much more biologically accurate phenomenology.

Although our work focused on characterizing hippocampal Schaffer collateral synapses, our approach can apply just as well to other synapse types. Additionally, other internal synaptic processes such as vesicle recycling may be combined with our model within an event-driven framework. In this way, future refinements of our model will have the ability to capture physiologically realistic phenomenology, trial-to-trial variability, asynchronicity, and internal synaptic dynamics for a wide variety of synapses at very low computational cost. It can help to uncover the contributions of single synapses or synaptic features to network computations, establishing a connection from molecular kinetics through synaptic phenomenology up to whole-network dynamics. This can be applied to create predictive models of biological networks from known physiological parameters or to design neuromorphic chips or spiking neural networks with dynamical properties relevant to future developments in brain-computer interfaces and artificial intelligence.

## Methods

### $Ca^{2+}$-evoked vesicle release model

The detailed model of molecular reaction-diffusion dynamics was developed in the modeling environment known as MCell [23,24,26]. The MCell model used as a basis for the design and validation of the presynaptic model presented in this paper comes from Nadkarni et al. [25]. It includes mechanisms for voltage-sensitive $Ca^{2+}$ influx and for $Ca^{2+}$ buffering in the presynaptic space, with pumps and channels in the membranes to maintain a steady-state average free $Ca^{2+}$ concentration of 100 nM [55].

When an action potential arrives at the presynaptic membrane, voltage-dependent $Ca^{2+}$ channels (VDCCs) open stochastically, traversing through four unopened states via voltage-dependent state transition rates [51] (see Fig A panel A in S1 Text), producing a $Ca^{2+}$ influx due to the steep electrochemical gradient [55]. The VDCCs very quickly shut off after the membrane potential returns to baseline (Fig B in S1 Text), and the newly introduced $Ca^{2+}$ ions diffuse randomly in the presynaptic space. Vesicle fusion occurs when a sufficient number of $Ca^{2+}$ ions have diffused over and bound to the release machinery associated with the SNARE complex of a docked vesicle [5,9,25]. Cytoplasmic calbindin (CB) with a concentration of 45 μM acts as a buffer that modulates the magnitude and duration of the free $Ca^{2+}$ (Fig A panel B in S1 Text) [58], and plasma membrane $Ca^{2+}$-ATPase (PMCA) pumps (Fig A panel C in S1 Text) actively remove $Ca^{2+}$ ions over a time course of seconds [59] to the baseline $[Ca^{2+}]_i$ of 100 nM [25,55]. Parameter values for these molecular mechanisms are given in Table A in S1 Text.

Calcium concentration as a function of time is measured at various spatial locations within the presynaptic space (see Fig 2 for synaptic structure and "Estimating $[Ca^{2+}]_i$ from Collision Events" in the Supplemental Information for a description of how local $[Ca^{2+}]$ is measured). For each spike train, local $[Ca^{2+}]_i$ in the vicinity of the SNARE complex was averaged over 2000 trials at a resolution of 0.1 msec. These calcium transient profiles were then used to drive deterministic simulations of SNARE state probabilities, as described below.

## Deterministic simulations of state probabilities

With the $Ca^{2+}$ profiles obtained for each spike-evoked simulation in MCell and with the state transitions for the release mechanisms defined in Fig 3A and 3B and Table 1, the corresponding vesicle release-rate profiles become computable. While the MCell simulations we ran do generate their own sets of release times, it would be impossible to fit precise phenomenological functions to the release histograms without running an infeasibly large number of trials. Furthermore, vesicle depletion following release events confounds the representation of release rate, its functional form, and its facilitation dynamics (see Fig 4). Therefore, instead of running millions of trials of MCell (or more) to produce temporally precise single-vesicle release rate histograms, we used the averaged calcium profiles from 2000 trials (at a resolution of 0.1 msec) to drive a deterministic simulation of the SNARE complex, tracking the probabilities of being in each state as functions of time. This approach, in effect, produced the average release histograms equivalent to an infinite number of trials acting on the averaged calcium traces.

This method tracked the probabilities of a particular release mechanism being in each possible state at every time step. That is, each state represents the number of $Ca^{2+}$ ions bound to the release molecule (0 through 5 for Syt-1 (synchronous; S) and 0 through 2 for Syt-7 (asynchronous; A)). Each mechanism $X \in \{S, A\}$ has a state probability vector $\mathbf{s}^X(t)$ associated with it that tracks the probability of being in each possible state, that is, the probability of having $n$ $Ca^{2+}$ ions currently bound for $n \in \{0 \ldots N_X\}$, where $N_S = 5$ and $N_A = 2$. State probabilities add to unity, and they update on each time step according to a $[Ca^{2+}]$-dependent state transition rate matrix $\mathbf{T}^X$, whose superdiagonal terms are the unbinding rates, moving from a higher to a lower-bound state, and whose subdiagonal terms are the binding rates, moving from a lower to a higher-bound state. Specifically, for mechanism $X \in \{S, A\}$ with $N_X \in \{5, 2\}$ calcium ions needed for release to occur, the binding rate is

$$T^X_{n+1,n} = (N_X - n) \cdot k_{X+} \cdot [Ca^{2+}], \tag{17}$$

and the unbinding rate is

$$T^X_{n-1,n} = n \cdot b^{n-1} \cdot k_{X-} \tag{18}$$

for $n \in \{0 \ldots N_X\}$ ions currently bound, where $b = 0.25$ acts as a binding cooperativity factor (see Fig 3 and Fig H in S1 Text for state transition diagrams and Table 1 for parameter values). Notice that we index the rows and columns of $\mathbf{T}^X$ (as well as the dimensions of $\mathbf{s}^X$) starting from 0 rather than 1 for convenience in representing the number of calcium ions bound in the current state (column) and in the next state (row). To enforce conservation of mass, the diagonal terms must equal the combined rate of leaving the current state through both binding and unbinding:

$$T^X_{n,n} = -(T^X_{n+1,n} + T^X_{n-1,n})$$

$$= -((N_X - n) \cdot k_{X+} \cdot [Ca^{2+}] + n \cdot b^{n-1} \cdot k_{X-}). \tag{19}$$

The above formulation does not yet take into account vesicle fusion. Recall that each mechanism $X$ induces vesicle release at a certain rate $\gamma_X$ from its releasable state (all $Ca^{2+}$ ions bound). Therefore, in addition to the unbinding rate, the diagonal term for the fully bound state of each mechanism also includes the rate of transition to the release state:

$$T^X_{N_X,N_X} = -(\gamma_X + N_X \cdot b^{N_X-1} \cdot k_{X-}), \tag{20}$$

where $\gamma_X$ is the mechanism-specific release rate defined in Table 1. By including the release

rate, the sum of state probabilities would slowly decay towards zero as probability mass leaks into the (untracked) release state, since when release occurs, the vesicle can no longer participate in further activity. This depletion of vesicle probability would obscure the true shape of the single-vesicle release rate. To account for this, the state vector is renormalized at each time step by the probability of no release event having yet occurred, such that the occupancies of each state again add to one:

$$\mathbf{s}^X(t) \leftarrow \frac{\mathbf{s}^X(t)}{\sum_{n=0}^{N_X} s_n^X(t)}. \tag{21}$$

If one considers the deterministic simulation to represent a state histogram averaged over an infinite number of trials, this normalization step effectively "zooms in" on the fraction of trials at each time step for which no release occurred. Thus, the model tracks the instantaneous rate of release, given that no release has yet occurred since the start of the simulation. This permits the calculation, for example, of the equilibrium state probability distribution (see Fig H in S1 Text, left pie charts), driven by the steady-state $Ca^{2+}$ concentration (100 nM in MCell: [25]). These equilibrium state vectors are essential for initializing all other simulations. From these, it is possible to determine the steady-state spontaneous release rates for each mechanism that result ($S_0 = 5.70 \times 10^{-9} ms^{-1}$; $A_0 = 1.84 \times 10^{-5} ms^{-1}$; see Fig C in S1 Text).

Because the matrix $\mathbf{T}^X$ represents transition rates rather than transition probabilities, it acts as the infinitesimal generator for a continuous-time finite state Markov process [90] rather than as a discrete transition matrix. Converting this exactly to a discrete transition probability matrix $\mathbf{P}^X$ using a time step of $\Delta t$ requires an infinite sum of matrix products, according to the Taylor series

$$\mathbf{P}^X\left([Ca^{2+}]; \Delta t\right) = \exp(\Delta t \cdot \mathbf{T}^X([Ca^{2+}])) = \sum_{m=0}^{\infty} \frac{(\Delta t \cdot \mathbf{T}^X([Ca^{2+}]))^m}{m!}. \tag{22}$$

This is akin to the probability of a Poisson process with transition rate $\lambda$ remaining in the same state for a time duration $\Delta t$, which follows the exponential form $p(t > \Delta t) = \exp(-\Delta t \cdot \lambda)$, acting just like the diagonal terms of the transition rate matrix and sharing the same Taylor series expansion. The time step $\Delta t$ needs to be small enough such that the level of $[Ca^{2+}]$ can be regarded as constant (on the order of 0.1 msec for the calcium transients investigated for this paper). However, in order to avoid too many matrix multiplications and sums, we chose a time step of 0.005 msec, which is small enough to use the linear approximation to Eq (22) while still maintaining numerical stability, even at very high levels of $[Ca^{2+}]$:

$$\mathbf{P}^X([Ca^{2+}]; \Delta t) \approx \mathbf{I} + \Delta t \cdot \mathbf{T}^X([Ca^{2+}]). \tag{23}$$

Thus, the state vector at time $t + \Delta t$ is the product of the state transition probability matrix with the state vector at time $t$, according to

$$\mathbf{s}^X(t + \Delta t) = \mathbf{P}^X([Ca^{2+}](t); \Delta t)\mathbf{s}^X(t)$$

$$= (\mathbf{I} + \Delta t \cdot \mathbf{T}^X([Ca^{2+}](t)))\mathbf{s}^X(t), \tag{24}$$

followed by the renormalization described above in Eq (21).

Similarly, the transitions in state probabilities for VDCCs, calbindin, and PMCA pumps in the well-mixed model were calculated according to

$$\mathbf{s}^M(t + \Delta t) = (\mathbf{I} + \Delta t \cdot \mathbf{T}^M)\mathbf{s}^M(t), \tag{25}$$

where $M \in \{VDCC, Cb, PMCA\}$ refers to the molecular species. Furthermore, the transition rate matrix $\mathbf{T}^{VDCC}$ is a function of membrane potential, while the matrices $\mathbf{T}^{Cb}$ and $\mathbf{T}^{PMCA}$ are functions of $[Ca^{2+}]_i$ (see Fig A and Table A in S1 Text for state transition diagrams and transition rate parameters). Within the well-mixed model simulations, diffusion occurs instantaneously, effectively eliminating space from consideration and allowing $[Ca^{2+}]_i$ to be the same for all molecular mechanisms within the synapse.

## Stimulus protocols for exploring facilitation

Whereas simulations with single action potentials can elucidate the functional form of synchronous and asynchronous release, stimulus trains of multiple spikes can reveal the dynamics of facilitation in release probability, which is well documented experimentally [14,44,52–54]. Short-term facilitation in release probability is more pronounced for spikes closer together in time than for those separated by long intervals. To investigate how delay affects probability of release, we studied paired-pulse facilitation (PPF) for interspike intervals (ISIs) of exponentially increasing delay. Specifically, we stimulated the MCell model with paired pulses of action-potential-like waveforms separated by 2, 5, 10, 20, 50, 100, and 200 ms and measured the local $[Ca^{2+}]_i$ at a point within the axon (see Fig 2 for synaptic structure and "Estimating $[Ca^{2+}]_i$ from collision events" in the Supplemental information for a description of how local $[Ca^{2+}]$ is measured). These $Ca^{2+}$ traces then drove deterministic simulations of synchronous and asynchronous release rate, as described in the previous section. This permitted us to determine a functional form to describe PPF (see Results).

Realistic spike trains, however, involve patterns much more complex than paired pulses, and the recent history of presynaptic activity can have a strong effect on future changes in release probability. To see how facilitation evolves in more complex trains of action potentials, we designed a protocol to explore a larger space of possible facilitated states, assuming that the level of facilitation experienced on one spike depends exclusively on the delay since the previous spike (the interspike interval, or ISI) and on the state of some internal facilitation parameter from the previous spike. The spike trains generally consist of two phases: a spiking ramp and a probe spike. The ramp phase explores how facilitation develops with multiple spikes at fixed ISIs and having anywhere from one to five spikes with an ISI of 2, 5, 10, or 20 ms (time prevented the exploration of ramps with more spikes). The probe phase explores how facilitation wears off with increasing delay between spikes and consists of a single spike at 2, 5, 10, 20, 50, 100, or 200 ms after the end of the ramp, as in the PPF protocol above. All these combinations of ramps and probes add up to 5×4×(7+1) = 160 cases (including those cases without a probe spike) or 136 unique spike trains (discounting the repeats with one spike in the ramp at different ISIs). Future work can explore the steady-state behavior of the facilitation factors by testing longer spike ramps of each ISI.

## Algorithms for fitting parameters and metaparameters

Fitting parameter values to the shapes of the release-rate histograms involved two steps: first, obtaining an initial guess, and second, optimizing the parameter values to a best-fit set. For the first step, the time constants for rate decay (see Eq (1) and (3) and Table 2) were found from the slopes of the logarithms of the profiles (see Eq (2) and Figs 5 and 6) in response to both $Ca^{2+}$ impulse and the $Ca^{2+}$ traces derived from MCell. Other parameters were initialized through trial and error. For the second step, we applied the Nelder-Mead simplex method of function optimization using the fminsearch() function in MATLAB 2016 [91] to minimize the cost function over the parameters. This method does not require a measure of the gradient of the cost function, which was not computable analytically. The cost function uses the fraction

of the variance unexplained (FVU) by the model, as in

$$\text{FVU}(y(t), f(t)) = \frac{\sum_{n=1}^{N_t} (y(t_n) - f(t_n))^2}{\sum_{n=1}^{N_t} (y(t_n) - \bar{y})^2},$$

(26)

where $\bar{y}$ is the mean release rate, $y(t)$ is the true shape of the release rate profile, and $f(t)$ is the model profile at the same $N_t$ time points. More precisely, the cost function $\epsilon$ is a linear combination of the FVU for the function and for the logarithm of the function:

$$\epsilon(y(t), f(t)) = \alpha \cdot \text{FVU}(y(t), f(t)) + \beta \cdot \text{FVU}(\log(y(t)), \log(f(t))),$$

(27)

where $\alpha$ and $\beta$ are constants ($\alpha = \beta = 1$ in the fits run for this paper). The FVU for the functions in linear space is more sensitive to the high-amplitude peaks that occur for the fast release components, while the FVU for the functions in logarithmic space is more sensitive to the slopes (and therefore the time constants) of the exponential components.

The metaparameters of the facilitation functions (see Eq (13) and (16) and Table 3) were fitted after the parameters were fitted to the release profiles in response to each spike of the trains described in the previous section of Methods. As described in Results, we took facilitation to apply only to the $P_c$ parameters, allowing them to vary within bounds for each spike in the train, while the profile time constants and the temporal filter parameters were held constant. The best-fit set of values for $P_c$ were found for the final spike-response profile of each spike train. The fitted parameters were taken as true, and the space of the logarithms of the metaparameters $\tau$, $N$, and $L = N^\xi$ was explored, using the same error function and optimization as above.

## Supporting information

**S1 Text. Supporting information in seven sections.** (A) Estimating $[Ca^{2+}]_i$ from collision events. (B) Chemical kinetics of calcium channels, buffers, and pumps. (C) Effects of buffer and spatial modeling on release dynamics. (D) Applying release start time filter to release rate profiles. (E) Facilitation nonlinearities. (F) Intuitive exploration of facilitation function behavior. (G) Goodness of fit of facilitation models. **Fig A. State Diagrams for VDCC, Calbindin, and PMCA.** All diagrams reproduced with permission from Nadkarni et al. [25]. (A) VDCC state transition model adapted from Bischofberger et al. [51]. Transition rates $\alpha_{ij}$ and $\beta_{ji}$ depend on membrane potential v. (B) State transitions for calbindin (CB) at high-affinity (H) and medium-affinity (M) $Ca^{2+}$-binding sites. On rates ($kh_+$ and $km_+$) are proportional to $[Ca^{2+}]_i$. (C) PMCA pump state diagram with $Ca^{2+}$ interactions depicted on the relative side of the membrane. $Ca^{2+}$ leakage occurs only in state $PMCA_0$. Association rate $kpm_1$ is proportional to $[Ca^{2+}]_i$. **Table A. Parameter Values for VDCC, Calbindin, and PMCA.** Table adapted from [25]. VDCC rates follow $\alpha_i(v) = \alpha_{i0}\exp(v/v_i)$ and $\beta_i(v) = \beta_{i0}\exp(-v/v_i)$. VDCC parameters values adapted from [51]. Calbindin parameter values adapted from [58]. PMCA parameter values adapted from [59]. **Fig B. Action-Potential-Evoked $Ca^{2+}$ Current.** (A) Action-potential-like waveform applied to axon. (B) Probability of a single VDCC being in the open state in response to the action potential in panel (A) increases from about $10^{-5}$ to around 96% during the spike before quickly shutting off; computed from deterministic simulation of state probabilities. (C) Rate of $Ca^{2+}$ influx through a single, pathologically open channel (red) and through a typical channel (blue), whose probability of being open follows (B). **Fig C. Spontaneous Rates of Vesicle Fusion Increase with $[Ca^{2+}]_{i0}$.** For small $[Ca^{2+}]_{i0}$, $S_0 = k_S \cdot ([Ca^{2+}]_{i0})^5$ and $A_0 = k_A \cdot ([Ca^{2+}]_{i0})^2$, where $k_S \approx 6 \times 10^{-4}$ ms$^{-1} \cdot \mu$M$^{-5}$ and $k_A \approx 2 \times 10^{-3}$ ms$^{-1} \cdot \mu$M$^{-2}$. As $[Ca^{2+}]_{i0} \longrightarrow \infty$, $S_0 \longrightarrow \gamma_S$ and $A_0 \longrightarrow \gamma_A$. Values for $S_0$ and $A_0$ at $[Ca^{2+}]_{i0} = 100$ nM, which is used throughout most of this paper, are pointed out for reference. **Fig D. Spatial Modeling**

**Important for Capturing Fine-Grain Features of Ca²⁺ Transients.** Color scheme identical to that used in Fig 2: yellow to blue represent proximal to distal Ca²⁺ sensors. (A) $[Ca^{2+}]_i$ measured at increasing distance from VDCC source (yellow to blue), with well-mixed approximation overlaid for comparison (maroon). Inset focuses on shorter time scale. (B) Profiles with calbindin removed from MCell (yellow to blue) and well-mixed model (maroon). Note that peak $[Ca^{2+}]_i$ for the most proximal case extends up to 81 μM, but is cut off for clarity. **Fig E. Effect of Calbindin Buffer on Spike-Evoked Ca²⁺ Profile and Release Rates.** Action-potential-like stimulus delivered to model axon starting at 0 ms. Diffusion is assumed to be instantaneous, and molecular state probabilities are tracked deterministically over time. (A) Free $[Ca^{2+}]_i$ with no calbindin buffer decays linearly with time due to saturation of PMCA pumps. (B) Syt-1/7-mediated release rates are large but short-lived in response to unbuffered Ca²⁺. (C) Free $[Ca^{2+}]_i$ with calbindin added to the axon has much smaller magnitude and much narrower peak but has much longer tail. (D) Vesicle release in response to buffered Ca²⁺ is much less pronounced. The calbindin buffer reduces the rate of synchronous transmission but extends the window for pronounced asynchronous transmission. **Fig F. Synchronous and Asynchronous Release Rates Decrease with Distance from the Ca²⁺ Source.** Color scheme identical to that used in Fig 2 and Fig D in S1 Text: yellow to blue represent proximal to distal Ca²⁺ sensors. (A) Synchronous release rate. (B) Integrated probability of synchronous release falls off nearly exponentially with distance to a baseline level. (C) Asynchronous release rates. (D) Integrated probability of asynchronous release also decays with distance to some baseline, but not exponentially. **Fig G. Parametric Fits to Release Histogram Profiles at Increasing Distance from the Ca²⁺ Source.** (A,B) Fitted release profiles (black) imposed over the true histograms for synchronous (A, blue) and asynchronous (B, red). (C) Parameter values as a function of distance for synchronous release. (D) The same for asynchronous release. **Fig H. Change in the Balance of Binding Kinetics and Internal State Distribution of Ca²⁺ Sensor with Spike History.** State diagrams the same as shown in Fig 3. (A) Synchronous state diagrams. At baseline $[Ca^{2+}]_i$ (first red dot), unbinding kinetics (left arrows) overpower binding (right arrows), biasing Syt-1 toward unbound state ($S_0$; top diagram), with almost no probability of having any Ca²⁺ ions bound before an action potential (left pie chart). During peak Ca²⁺ influx (second red dot), binding rates (thicker right arrows) overpower unbinding, biasing Syt-1 toward its fully-bound releasable state ($S_5$; lower diagram), with much greater probability of having at least some Ca²⁺ bound (right pie chart). (B) The same for asynchronous release with Syt-7, whose releasable state requires two Ca²⁺ ions bound ($A_2$). Slower kinetics lead to only slight bias in favor of binding during an action potential (slightly thicker right arrows in lower diagram), leading to miniscule increase in probability of being in the releasable state on later spikes (right pie chart). Release becomes more probable on subsequent spikes because previous activity has pushed synaptotagmin into higher-bound states, making reaching the releasable state easier. **Fig I. Empirical Facilitation in Release Probability is a Nonlinear Function of Spike History and Ca²⁺ Buildup.** (A) $[Ca^{2+}]_i$ and (B) release rate in response to a 5-spike ramp stimulus with a 10-ms ISI (black and dark green), followed by a single probe spike at increasing delay from the end of the ramp (gray and light green; multiple cases overlaid on the same plot). Release rate grows much faster than Ca²⁺ buildup can account for. **Fig J. Empirical Facilitation in Release Probability is a Nonlinear Function of Spike History.** Integrated release fidelity (P(n)) relative to baseline (P(0)) for the various stimulus cases explored. Ramp # indicates the number of spikes in the ramp preceding the probe spike, and Δt represents the ISI between the last ramp spike and the probe spike. Spike history noticeably affects the growth of facilitation, as seen for ramps with 2-ms ISIs (A), 5-ms ISIs (B), 10-ms ISIs (C), and 20-ms ISIs (D). Different colors distinguish facilitation functions with different spike histories. Dark lines follow relative release fidelity for spikes along spike ramps, and

dotted lines follow relative release fidelity for probe spikes. **Fig K. Saturation of Facilitation Parameters.** (A) Facilitation parameter f(·) increases almost linearly from one spike (f(n-1)) to the next (f(n)), until it approaches some limit N≥1. (B) Curves represent the unseen change in f(·) between spikes. Dots represent actual values observed at spike times, values determined by the $Ca^{2+}$-triggered increment in release fidelity at each spike. Steady-state value for facilitation parameter limited by stimulus frequency and by value of N. No facilitation above baseline occurs for N = 1. **Fig L. Release Rate Parameters and Facilitation Metaparameters Fitted to Empirical Histogram Profiles.** Errors across all cases in linear and logarithmic space for the predictive model.

(DOCX)

## Acknowledgments

We would like to thank Donald Spencer and Margot Wagner for all the perspectives, insights, and suggestions they offered as we prepared this manuscript for publication.

## Author Contributions

**Conceptualization:** Jonathan W. Garcia, Thomas M. Bartol.

**Data curation:** Jonathan W. Garcia.

**Formal analysis:** Jonathan W. Garcia.

**Funding acquisition:** Terrence J. Sejnowski.

**Investigation:** Jonathan W. Garcia, Thomas M. Bartol.

**Methodology:** Jonathan W. Garcia, Thomas M. Bartol.

**Project administration:** Terrence J. Sejnowski.

**Resources:** Thomas M. Bartol, Terrence J. Sejnowski.

**Software:** Jonathan W. Garcia, Thomas M. Bartol.

**Supervision:** Terrence J. Sejnowski.

**Validation:** Jonathan W. Garcia, Thomas M. Bartol, Terrence J. Sejnowski.

**Visualization:** Jonathan W. Garcia.

**Writing – original draft:** Jonathan W. Garcia.

**Writing – review & editing:** Jonathan W. Garcia, Thomas M. Bartol, Terrence J. Sejnowski.

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
