## [Decision Letter · Decision Letter 0]

12 Oct 2021

Dear Dr. Garcia,

Thank you very much for submitting your manuscript "Multiscale modeling of presynaptic dynamics from molecular to mesoscale" for consideration at PLOS Computational Biology. As with all papers reviewed by the journal, your manuscript was reviewed by members of the editorial board and by several independent reviewers. The reviewers appreciated the attention to an important topic. Based on the reviews, we are likely to accept this manuscript for publication, providing that you modify the manuscript according to the review recommendations.

Sincerely,

Joanna Jędrzejewska-Szmek, Ph.D.

Associate Editor

PLOS Computational Biology

Kim Blackwell

Deputy Editor

PLOS Computational Biology

[LINK]

Reviewer's Responses to Questions

**Comments to the Authors:**

Reviewer #1: The study by Garcia et al. is an important contribution that makes it possible to scale-up the implication of synaptic designs to networks of neurons. They have chosen an important synapse, the mossy fiber synapse in the trisynaptic circuit of the hippocampus for their modeling investigation. There exists thousands of morphologically distinct synapses that play a diverse roles in information transfer across neurons. The suggests a strong causal link between synaptic morphology and function. Yet, there is often a disconnect between understanding at the level of synapses and neuronal circuit dynamics in both experimental studies and modeling studies. On the other hand it is unrealistic to simulate in great spatial and physiological detail activity at the synapses when computing activity of hundreds of neurons. This manuscript attempts to do fill this gap.

I think, over-all results of the phenomenological are a good quantitative fit to detailed spatiotemporal calculations and therefore useful for scaling up the numbers of synapses in simulations and computation with neuronal networks. I do have a few concerns regarding the specifics of the model that are listed below.

• I see that various train stimuli have been explored, however it be useful to show comparison of the phenomeological model with realistic model for more realistic trains of activity (variable frequency and noisy stimulus). Since the parameters of the phenomenological models are retro-fit to specific data, how do the authors envision incorporating it in a network (since thats the point of this study) which sees time varying stimulus?

• Figure 9. Description of this figure in the results and and figure both are hard to understand and need to be expanded.

• I did not see a description of the mossy fiber synapse design at all. This is a surprising omission given that the model is of that of MF. For that matter, how is the model different from a schaffer collateral? What specifics of the MF have been included (RRP size, number of release sites etc.) MFs show distinct short-term plasticity profiles as described in detail by several papers from Peter Jonas’s lab, Katalin Toth lab and Joachim Lu?bke lab. This needs to be included

Minor comments

• References list seems really long, is a more concise list possible?

• It will be useful if the model is available on a public repository like a modelDB. Also final parameters (taus of decay and facilitation etc.) of the model that fit the detailed MCell simulations for various stimuli could be presented in tabular format for ease of use.

Reviewer #2: Evaluation of the manuscript by J.W. Garcia, T.M. Bartol and T.J. Sejnowski entitled: Multiscale modeling of presynaptic dynamics from molecular to mesoscale

Comments

This study is an attempt to develop a presynaptic model of processes determining the dynamics of transmission efficacy. First, the traces of free Ca2+ concentration were obtained that occur in response to various action potential stimulus trains from a Monte Carlo model of a hippocampal mossy fiber axon. Ca2+ was estimated as it diffuses from the voltage dependent Ca channel (VDCC) source, through the buffer, to the Ca2+ sensor in the SNARE complex and its time course determined at varying distance from a VDCC cluster with Ca2+ being buffered by calbindin. The calcium traces then drive deterministic state vector models of synaptotagmin 1 and 7 (Syt-1/7), which respectively mediate synchronous and asynchronous release in excitatory hippocampal synapses. This paper extends their earlier Monte Carlo studies of synaptic transmission, and it comes from a prominent group that worked on these and related problems for a long time. The methodology is well described and discussed. Moreover, there is a definitive need for simulations that combine Ca dynamics and release processes.

Authors stress the advantages of this approach over biological experiments in terms of both flexibility and precision when it comes to model validation are discussed. They also discuss disadvantages – a) Does Markov chain Monte Carlo framework accurately replicate the dynamics of diffusion and molecular interactions of these mechanisms? b) Is the molecular kinetics of the included species their true kinetics? and c) Are the molecular species and processes included the only relevant ones? That is needed and appropriate.

Criticisms

I have two basic issues with this paper:

First issue relates to Ca diffusion and how it is addressed. As authors state diffusion plays a key role in presynaptic processes. However, the question of how micro or nanostructures like cytoskeleton affect diffusion and ultimately release is just briefly mentioned. Why? Is it so time consuming to do some additional simulations with idealized geometry to answer the question how such intracellular micro-structures affect the diffusion and release? The importance of vesicles themselves (or of vesicular clusters) as diffusional barriers is not addressed at all. It should be. It is well documented that they play an important role in restricting diffusion. See work of Cooper et al., (1996) J Neurophysiol 75:2451; Glavinovic & Rabie, Pflügers Arch (2001) 443:132; Shahrezaei et al., (2006) J. Neurosci.26:13240. In central synapses individual vesicles play a less important role as diffusion barriers than in adrenal secretory cells, but they cluster. Doing additional simulations with vesicles included would be ideal, but even in the absence of new simulations it should be discussed, earlier work referred to and commented upon. Authors discuss local saturation of both the calbindin buffer and the PMCA pumps in the nanodomains near the VDCC cluster, where free Ca2+ reaches high levels and how it affects proximal Ca curve. Please expand on this in the context of slower diffusion caused by vesicular presence.

Second, this paper presents a model for hippocampal mossy fiber axon synapse with a given set of Ca binding-unbinding parameters. However, these parameters which determine the total binding ratio (the ratio of bound to free Ca2+) varies greatly in different secretory systems (Stuenkel (1994) J Physiol 481:251; Tank et al. (1995) J Neurosci 15:7940). Moreover, the binding ratio changes with cell development because the expression of Ca2+ buffers is developmentally regulated (Friauf (1993) J Comp Neurol 334:59; Lohman and Friauf (1996) J Comp Neurol 367:90. How would different Ca binding-unbinding parameters change Ca dynamics and release processes? Could authors comment on these issues, and if possible, do some simulations to answer them more completely. These changes would make this work significantly more interesting to a wider audience.

**Have the authors made all data and (if applicable) computational code underlying the findings in their manuscript fully available?**

Reviewer #1: **No: **

Reviewer #2: Yes

PLOS authors have the option to publish the peer review history of their article (what does this mean?). If published, this will include your full peer review and any attached files.

Reviewer #1: No

Reviewer #2: No

Figure Files:

Data Requirements:

Reproducibility:

References:

If you need to cite a retracted article, indicate the article’s retracted status in the References list and also include a citation and full reference for the retraction notice.

---

## [Decision Letter · Decision Letter 1]

29 Mar 2022

Dear Dr. Garcia,

We are pleased to inform you that your manuscript 'Multiscale modeling of presynaptic dynamics from molecular to mesoscale' has been provisionally accepted for publication in PLOS Computational Biology.

Best regards,

Joanna Jędrzejewska-Szmek, Ph.D.

Associate Editor

PLOS Computational Biology

Kim Blackwell

Deputy Editor

PLOS Computational Biology

Reviewer's Responses to Questions

**Comments to the Authors:**

Reviewer #1: My concerns have been addressed in this version of the manuscript. I think the manuscript is ready to be published at this point.

Reviewer #2: This is an excellent paper by a very competent group. It was a pleasure to read it. Congratulations.

**Have the authors made all data and (if applicable) computational code underlying the findings in their manuscript fully available?**

Reviewer #1: Yes

Reviewer #2: Yes

PLOS authors have the option to publish the peer review history of their article (what does this mean?). If published, this will include your full peer review and any attached files.

Reviewer #1: No

Reviewer #2: No

---

## [Editor Report · Acceptance letter]

1 May 2022

PCOMPBIOL-D-21-01582R1 

Multiscale modeling of presynaptic dynamics from molecular to mesoscale

Dear Dr Garcia,

I am pleased to inform you that your manuscript has been formally accepted for publication in PLOS Computational Biology. Your manuscript is now with our production department and you will be notified of the publication date in due course.

With kind regards,

Andrea Szabo
